# KᴀLM-Eᴍʙᴇᴅᴅɪɴɢ-V2: Sᴜᴘᴇʀɪᴏʀ Tʀᴀɪɴɪɴɢ Tᴇᴄʜɴɪǫᴜᴇs ᴀɴᴅ Dᴀᴛᴀ Iɴsᴘɪʀᴇ A Vᴇʀsᴀᴛɪʟᴇ Eᴍʙᴇᴅᴅɪɴɢ Mᴏᴅᴇʟ

**Xinping Zhao**[1*], **Xinshuo Hu**[2*], **Zifei Shan**[2], **Shouzheng Huang**[1], **Yao Zhou**[2],
**Xin Zhang, Zetian Sun, Zhenyu Liu, Dongfang Li, Xinyuan Wei, Youcheng Pan,**
**Yang Xiang, Meishan Zhang, Haofen Wang**[3]**, Jun Yu, Baotian Hu**[1†]**, Min Zhang**[1]
[1]Shenzhen Loop Area Institute (SLAI); [2]Tencent, Shenzhen, China; [3]Tongji University
xinpingzhao@slai.edu.cn, xinshuohu@tencent.com
shouzhenghuang912@gmail.com, yoozhou@tencent.com
zifeishan@tencent.com, {baotianhu, minzhang}@slai.edu.cn
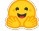 https://huggingface.co/KaLM-Embedding

## Aʙsᴛʀᴀᴄᴛ

Recent advancements in Large Language Models (LLMs)-based text embedding models primarily focus on data scaling or synthesis, yet limited exploration of training techniques and data quality, thereby constraining performance. In this work, we propose KaLM-Embedding-V2 from the Lychee-KaLM team, a series of versatile and compact embedding models, systematically incentivizing advanced embedding capability in LLMs by superior training techniques and high-quality data. For model architecture, we implement the models on a 0.5B compact size with simple mean-pooling to produce fixed-length embeddings and remove the causal attention mask to enable fully bidirectional representation learning. For training techniques, we propose a progressive multi-stage training pipeline: pre-training on weakly supervised large-scale datasets, fine-tuning with supervised high-quality datasets, and contrastive distillation with fine-grained soft signals, integrated with focal-style reweighting and online hard-negative mixing to emphasize difficult samples and enrich hard negatives, respectively. For training data, we curate over 20 categories for pre-training and 100 categories for fine-tuning and contrastive distillation, to improve both performance and generalization, leveraging task-specific instructions, hard-negative mining, and example-based multi-class labeling to ensure high quality. Combining these techniques, our KaLM-Embedding-V2 series achieves state-of-the-art performance on the Massive Text Embedding Benchmark, outperforming models of comparable size and rivaling models 3–26x larger, setting a new standard for versatile and compact embedding models under 1B parameters. The code, data, and models are available at https://kalm-embedding.github.io/.

## 1 Iɴᴛʀᴏᴅᴜᴄᴛɪᴏɴ

Text embedding encapsulates text semantics and serves as fundamental infrastructure in numerous natural language processing (NLP) tasks (Muennighoff et al., 2023a; Xiao et al., 2024), including retrieval (Nguyen et al., 2016), reranking (Liu et al., 2018b), classification (McAuley & Leskovec, 2013), and semantic textual similarity (STS) (Agirre et al., 2012), etc. Recently, retrieval-augmented generation (RAG) has gained increasing attention in LLMs (Wu et al., 2022; Gao et al., 2023; Huang & Huang, 2024; Zhao et al., 2024; 2025; Rao et al., 2025; Chen et al., 2025a), where embedding models play a crucial role in RAG. It enables the efficient retrieval of external information to complement LLMs' outdated, incomplete, or inaccurate internal knowledge. With the advancement of LLMs, embedding models have become the primary bottleneck for improvement within the

---

[*]Equal contribution
[†]Corresponding Author

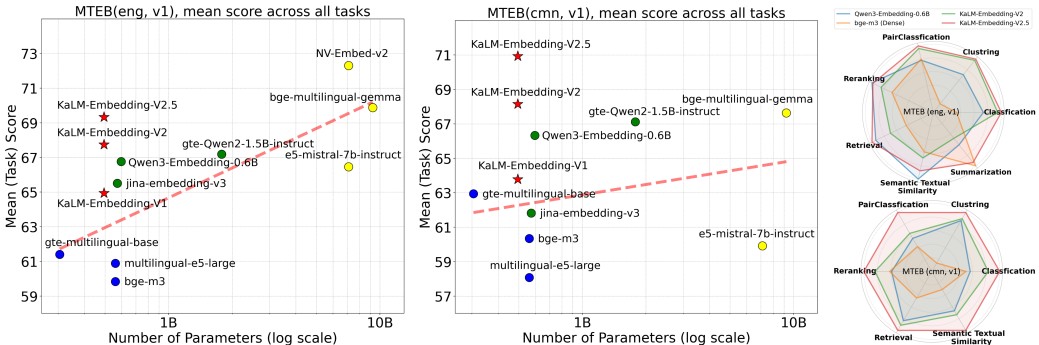

Figure 1: (**Left**) Comparison between the `KaLM-Embedding` series and other models on MTEB. The red dashed line depicts the logarithmic trendline fitted to the performance data of all the baseline models. The colors represent models with comparable parameter scales, with each group of models sharing the same parameter scale assigned a consistent color. (**Right**) Radar charts show our models achieve SOTA performance in a wide array of tasks.

RAG framework (Setty et al., 2024), which leads to the emergence of numerous text embedding models (Zhang et al., 2025c; Lee et al., 2025a;b;b; 2024; Huang et al., 2024; Xiao et al., 2024; Li et al., 2023; Vera et al., 2025).

Although numerous text embedding models have been built on massive or synthetic data (Zhang et al., 2025c; Lee et al., 2025b; 2024), they fall short in exploring superior training techniques and high-quality data, as well as how different training techniques, architecture designs, and data curation strategies can be systematically orchestrated to maximize the full potential of embedding capabilities in LLMs. Furthermore, most state-of-the-art (SOTA) embedding models originate from industry, where proprietary data, closed training code, commercial restrictions, and limited reproducibility pose challenges for academic research. To this end, it is necessary and valuable to establish new standards for open-source embedding models, emphasizing versatility and compactness—two crucial properties demanded in real-world scenarios where accuracy and efficiency are paramount. By fully open-sourcing models, code, and data with commercial use permitted, we aim to ensure transparency and reproducibility, thereby facilitating academic research and enabling widespread practical applications.

In this work, we propose `KaLM-Embedding-V2`, a series of versatile and compact general-purpose text embedding models, enhanced with the well-designed model architecture, superior training techniques, and high-quality data curation, which aim to incentivize advanced **K**nowledge in l**a**rge **L**anguage **M**odels into **Embedding** Models. Specifically, we make the following four innovations:

- For model architecture, our `KaLM-Embedding-V2` series are implemented upon a 0.5B compact size, with a simple yet effective mean-pooling layer to produce fixed-length embeddings. To further improve representation learning, we remove the causal attention mask of decoder-only LLMs and enable bidirectional attention during training as well as inference, which has been proven to be more effective for representation learning (Lee et al., 2025a;b; Sturua et al., 2024; Li et al., 2023).

- For training recipe, we implement a progressive multi-stage training pipeline, starting with the Qwen2-0.5B (Yang et al., 2024). Specifically, the training begins with pre-training on large-scale weakly supervised datasets that may include noise, then fine-tuning on relatively smaller, high-quality, supervised datasets, followed by contrastive distillation on fine-grained soft signals that capture nuanced differences. The multi-stage training pipeline progressively incentivizes advanced embedding capabilities in LLMs from coarse-grained to fine-grained representation learning.

- For training objective, previous works (Lee et al., 2025a; Hu et al., 2025) equally treat each training sample, making the optimization direction dominated by the majority of easy samples. Inspired by (Lin et al., 2017), we introduce a focal-style reweighting mechanism to emphasize difficult samples. However, as training progresses, offline mined hard negatives become less challenging. To provide continual informative hard negatives, we propose synthesizing new hard ones via online pair-wise or list-wise mixing. Unlike offline mining, our online hard negative mixing blends features of existing hard negatives to generate new ones, significantly reducing computational cost.

- For training data, we curate over 20 categories of data for pre-training and 100 categories of data for fine-tuning and distillation. We present a comprehensive recipe for curating high-quality training data, including dataset-specific construction, task-specific instructions, hard-negative mining, and

example-based multi-class labeling. This allows the research community to reproduce the model and considerably lowers the entry barrier, facilitating the development of embedding models.

Combining these innovative techniques, our `KaLM-Embedding-V2` series obtains impressive performance on the Massive Text Embedding Benchmark (MTEB) English (eng) (Muennighoff et al., 2023a) and Chinese (cmn) (Xiao et al., 2024), significantly outperforming models of comparable size, as shown in Figure 1. Remarkably, even at a 0.5B size, the `KaLM-Embedding-V2` series competes with 3–26× larger models. Out-of-domain (OOD) evaluation (Appendix C), matryoshka embedding evaluation (Appendix D), case study (Appendix E), visualization analysis (Appendix F), and multilingual evaluation (Appendix G) are provided in Appendices due to the page limit. In a nutshell, the proposed model exhibits strong OOD generalization, competing with the 15x larger model in real-world retrieval scenarios; it maintains robust performance with matryoshka embeddings even at smaller dimensions, *e.g.,* 256; case studies show its enhanced discriminative capacity in distinguishing positive passages from hard negatives; visualization analysis reveals superior intra-class compactness and inter-class separability clusters; and multilingual evaluation show that its performance is comparable to SOTA multilingual embedding models, even though it was not trained on large-scale multilingual corpora.

## 2 RELATED WORK

**Text embedding models.** Text embeddings (Zhang et al., 2025b), which are vectors encapsulating text semantics, are fundamental for NLP tasks such as retrieval (Nguyen et al., 2016), reranking (Liu et al., 2018b), and classification (McAuley & Leskovec, 2013). BERT (Devlin et al., 2018) marked a significant milestone, using masked language modeling to pre-train deep bidirectional Transformer encoders for powerful contextual modeling. A breakthrough for sentence similarity tasks was Sentence-BERT (SBERT) (Reimers & Gurevych, 2019), which fine-tuned BERT-like models with query-passage pairs to generate semantically meaningful sentence embeddings directly comparable via similarity. Another prominent example is the Text-to-Text Transfer Transformer (T5) (Raffel et al., 2019) which follows a fully encoder-decoder architecture and reframes all NLP tasks as text-to-text generation. While not initially designed for text embedding, the encoder portion of T5 can be used to generate powerful sentence representations. To systematically assess the robustness, generalization, and task-transferability of such embedding models, comprehensive benchmarks like the Massive Text Embedding Benchmark (MTEB) (Muennighoff et al., 2023a; Xiao et al., 2024; Enevoldsen et al., 2025) have emerged. These benchmarks provide critical insight into how well embedding models perform in real-world, diverse scenarios, driving further research in text embedding.

**LLMs as embedding models.** Pioneering studies explored the feasibility of leveraging LLMs for representation learning by adapting generative or encoder-decoder architectures into embedding models. E5 (Wang et al., 2022) unified retrieval, classification, and NLI tasks under a contrastive framework using in-batch negatives. GTR (Ni et al., 2022) fine-tuned T5 models for dual-encoder retrieval tasks. INSTRUCTOR (Su et al., 2023) introduced instruction tuning for embeddings, enabling task-specific representation via natural language prompts. Recently, LLMs, characterized by their massive scale and remarkable capacity, have become a prevailing paradigm in generating high-quality text embeddings. Many embedding models using LLMs as the backbone, *e.g.,* BGE (Li et al., 2025), NV-Emb (Lee et al., 2025a), E5-Mistral (Wang et al., 2024a), GTE (Li et al., 2023; Zhang et al., 2025c), Jina (Sturua et al., 2024; Akram et al., 2026), as well as (Hu et al., 2025), mainly initialized from the Mistral or Qwen, etc, have achieved substantial improvements over earlier encoder-based models such as BERT and T5. Adapting LLMs into embedding models requires sophisticated training strategies, *e.g.,* contrastive pre-training to draw semantically similar inputs together (Gao et al., 2021), instruction tuning to tailor embeddings for downstream tasks (Su et al., 2023), contrastive distillation for compression (Rao et al., 2023; Zhang et al., 2025a; Akram et al., 2026), and hard-negative mining to enforce fine-grained distinctions. Although studied for ages, systematic research of superior training techniques and high-quality data curation is still underexplored.

## 3 METHOD

In this section, we present comprehensive technical details of the `KaLM-Embedding-V2` series, including model architecture designs, training objectives, training recipes, and data curation strategies.

Figure 2: The overall training workflow of the `KaLM-Embedding-V2` series. The left illustrates the workflow of contrastive learning, while the right shows that of contrastive distillation.

## 3.1 MODEL ARCHITECTURE

The `KaLM-Embedding-V2` series is initialized from Qwen2-0.5B (Yang et al., 2024) and further tuned, which enables our embedding models to leverage the vast knowledge already encoded in its parameters. While causal attention masks are commonly used in LLMs for language modeling, they are not well-suited for representation learning, thereby hindering embedding capacity (Lee et al., 2025a;b; Sturua et al., 2024; Li et al., 2023). To address this, we remove the causal attention mask and enable fully bidirectional attention. For text embedding, an input sequence $\mathcal{T}$ of length $L$ is processed by `KaLM-Embedding-V2`, denoted as $\mathcal{K}(\cdot)$, to produce token embeddings $\mathbf{T}_{\text{emb}} \in \mathbb{R}^{L \times d}$. A pooling layer $\mathcal{P}(\cdot)$ is then applied to obtain a single embedding $\mathbf{E} \in \mathbb{R}^d$ representing the entire input:

$$\mathbf{T}_{\text{emb}} = \mathcal{K}(\mathcal{T}), \quad \mathbf{E} = \mathcal{P}(\mathbf{T}_{\text{emb}}), \tag{1}$$

where $d$ is the hidden dimension. Following prior works (Lee et al., 2025a;b; Hu et al., 2025), we set $\mathcal{P}(\cdot)$ as the simple yet effective mean pooling. The input $\mathcal{T}$ consists of the task instruction (optional) and the query/passage, as described in §3.4. The overall training workflow is illustrated in Figure 2.

## 3.2 TRAINING OBJECTIVE

**Contrastive Learning.** The `KaLM-Embedding-V2` series was mainly trained with the contrastive loss, specifically InfoNCE (Gutmann & Hyvärinen, 2010), which maximizes the agreement of positive pairs while minimizing that of negative pairs. The workflow of contrastive learning is illustrated on the left side of Figure 2. Generally, a training batch is organized as $\{I_i, q_i, p_i^+, p_{i,1}^-, p_{i,2}^-, ..., p_{i,M}^-\}_{i=0}^N$, where $N$ is the batch size. Each sample consists of a task instruction $I_i$, a query $q_i$, a positive target $p_i^+$, and (optionally) $M$ hard negatives $\{p_{i,1}^-, p_{i,2}^-, \ldots, p_{i,M}^-\}$. Before loss computation, the query $q_i$ and passages ($p_i^+$ and $p_{i,*}^-$) are encoded as vectors:

$$\mathbf{q}_i = \mathcal{P}(\mathcal{K}(I_i \oplus q_i)), \quad \mathbf{p}_i^+ = \mathcal{P}(\mathcal{K}(p_i^+)), \quad \mathbf{p}_{i,*}^- = \mathcal{P}(\mathcal{K}(p_{i,*}^-)), \tag{2}$$

where $\oplus$ denotes concatenation. For most tasks, the instruction is prepended only to the query, while for symmetric tasks, it is also prepended to the passages, as detailed in Table 1. Having established the embedding vectors of queries, positive targets, and hard negatives, for each mini-batch of size $N$, we optimize the contrastive learning objective with in-batch negatives and in-batch hard negatives as:

$$\mathcal{L} = \mathop{\mathbb{E}}_{i \in N} \left[ -\log \frac{e^{s(\mathbf{q}_i, \mathbf{p}_i^+)/\tau}}{Z_i} \right], \quad Z_i = e^{s(\mathbf{q}_i, \mathbf{p}_i^+)/\tau} + \sum_{j \neq i}^N e^{s(\mathbf{q}_i, \mathbf{p}_j^+)/\tau} + \sum_j^N \sum_k^M e^{s(\mathbf{q}_i, \mathbf{p}_{j,k}^-)/\tau}, \tag{3}$$

where $s(\cdot)$ measures the similarity between two embedding vectors, which is set as the cosine similarity function; $\tau$ is the temperature coefficient; the three terms in the denominator $Z_i$ represent (1) the positive target, (2) in-batch negatives, and (3) in-batch hard negatives, respectively.

**Focal-style Reweighting Mechanism.** While effective, the above training objective treats each sample equally, making the optimization direction dominated by the majority of easy samples. Inspired by (Lin et al., 2017), we re-weight each sample according to its difficulty, where the more difficult the sample, the larger the weight, thereby focusing on learning difficult samples. The loss weight and the optimized training objective are defined as follows:

$$w_i = (1 - \frac{e^{s(\mathbf{q}_i, \mathbf{p}_i^+)/\tau}}{Z_i})^\gamma, \quad \mathcal{L} = \mathop{\mathbb{E}}_{i \in N} \left[ -w_i \log \frac{e^{s(\mathbf{q}_i, \mathbf{p}_i^+)/\tau}}{Z_i} \right], \tag{4}$$

where $\gamma \in [0, +\infty)$ is a focusing parameter controlling the skewness of the weighting scheme. When $\gamma = 0$, the objective reduces to the standard form with uniform weighting. As $\gamma$ increases, the loss pays more attention to the difficult samples than the easy ones.

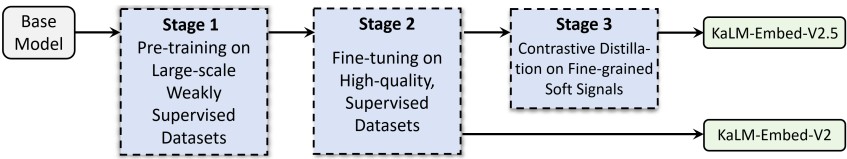

Figure 3: Multi-stage training pipeline of the `KaLM-Embedding-V2` series.

**Online Hard Negative Mixing Strategy.** As training progresses, offline mined hard negatives become less difficult after several training iterations. To provide continual informative hard negatives throughout the training, previous works typically re-mines hard negatives after every fixed number of steps (*e.g.*, 1000), which largely reduces training efficiency. To this end, we propose an online hard negative mixing strategy that synthesizes new informative hard negatives via pair-wise/list-wise mixing, in favor of effectiveness and efficiency. The pair-wise/list-wise mixing can be formulated as:

$$\mathbf{h}_i^- = \frac{\tilde{\mathbf{h}}_i^-}{\|\tilde{\mathbf{h}}_i^-\|_2}, \quad \tilde{\mathbf{h}}_i^- = \lambda \mathbf{p}_{i,j}^- + (1-\lambda)\mathbf{p}_{i,k}^-, \quad j \neq k, \quad j,k \in [1,M] \tag{5}$$

$$\mathbf{s}_i^- = \frac{\tilde{\mathbf{s}}_i^-}{\|\tilde{\mathbf{s}}_i^-\|_2}, \quad \tilde{\mathbf{s}}_i^- = \sum_{m=1}^{M} \lambda_m \mathbf{p}_{i,m}^-, \quad \text{s.t.} \sum_{m=1}^{M} \lambda_m = 1, \tag{6}$$

where $\mathbf{h}_i^-$ and $\mathbf{s}_i^-$ denote pair-wise and list-wise synthetic hard negatives, respectively; $\|\cdot\|$ is the $l_2$-norm; $\mathbf{p}_{i,j}^-$ and $\mathbf{p}_{i,k}^-$ are randomly drawn from the hard negative set $\{p_{i,1}^-, \ldots, p_{i,M}^-\}$ without replacement; $\lambda \sim \text{Beta}(\alpha = 2, \beta = 2)$, $\lambda \in (0,1)$; and $\lambda_m = e^{s(\mathbf{q}_i, \mathbf{p}_{i,m}^-)} / \sum_j^M e^{s(\mathbf{q}_i, \mathbf{p}_{i,j}^-)}$. The mixing incurs negligible overhead. After synthesis, $\mathbf{h}_i^-$ and $\mathbf{s}_i^-$ are incorporated into the denominator $Z_i$ as additional hard negatives for query $q_i$:

$$\mathcal{Z}_i = Z_i + \sum_j^N e^{s(\mathbf{q}_i, \mathbf{h}_j^-)/\tau} + \sum_j^N e^{s(\mathbf{q}_i, \mathbf{s}_j^-)/\tau}, \quad \mathcal{L} = \mathbb{E}_{i \in N}\left[-w_i \log \frac{e^{s(\mathbf{q}_i, \mathbf{p}_i^+)/\tau}}{\mathcal{Z}_i}\right], \tag{7}$$

where multiple synthetic negatives can be applied, though only one is illustrated here for clarity.

**Contrastive Distillation.** Unlike previous works trained solely with coarse-grained hard signals, we further perform contrastive distillation by distilling fine-grained soft signals, *i.e.,* the normalized distribution of temperature-scaled cosine similarity scores from a stronger teacher model (Qwen3-Embedding-8B (Zhang et al., 2025c)). This encourages the embedding model to capture nuanced differences between the positive and negative. Specifically, the training objective minimizes the discrepancy between the teacher's and the student's distributions. Formally, following (Hinton et al., 2015), we employ the Kullback–Leibler (KL) divergence as the contrastive distillation objective:

$$\mathcal{L}_{KL} = D_{KL}(P_t \| P_s) = \sum_i P_t(i) \log \frac{P_t(i)}{P_s(i)}, \quad P_t(i) = \frac{e^{z_{t,i}/\tau}}{\sum_j e^{z_{t,j}/\tau}}, \quad P_s(i) = \frac{e^{z_{s,i}/\tau}}{\sum_j e^{z_{s,j}/\tau}} \tag{8}$$

where $P_t$ and $P_s$ represent the teacher's and student's distribution of similarity scores, respectively; $P_t(i)$ and $P_s(i)$ denote the $i$-th entry; $z_{*,i}$ represents the $i$-th similarity score. We find that continual training with contrastive distillation yields substantial improvements over further fine-tuning with contrastive learning. It is worth mentioning that the proposed contrastive distillation is model-agnostic and can be applied to any embedding models. The working flow of contrastive distillation is shown on the right of Figure 2.

**Matryoshka Representation Learning (MRL).** We incorporate MRL (Kusupati et al., 2022) into both the contrastive (Equation 7) and KL loss (Equation 8) to enable flexible-dimensional embeddings, which leads to the best overall performance with matryoshka embeddings as shown in Appendix D.

### 3.3 TRAINING RECIPE

To progressively incentivize embedding capabilities in LLMs, we introduce a multi-stage training pipeline that smoothly transitions from coarse-grained to fine-grained representation learning: (1) Pre-training, (2) Fine-tuning, and (3) Contrastive distillation, as described below.

Table 1: The task instruction of query for training and evaluation.

| | Task Type | Instruction | Example |
|---|---|---|---|
| Asymmetric | Retrieval, Reranking | General | Instruct: Given a query, retrieve documents that answer the query. \n Query: {query} |
| | Classification, Clustering | Specific | Instruct: Categorizing the given news title \n Query: {query} |
| Symmetric | STS, Pair Classification | General | Instruct: Retrieve semantically similar text. \n Query: {query} |

**Pre-training.** The `KaLM-Embedding-V2` series is first pre-trained on large-scale, weakly supervised datasets spanning over 20 categories (refer to Table 16 for details) to learn general-purpose representations. This stage employs the training objective in Equation 3, using only in-batch negatives. The comprehensive pre-training endows the model with strong generalization.

**Fine-tuning.** Next, the model is fine-tuned on over 100 categories of high-quality supervised datasets covering both retrieval and non-retrieval tasks, such as STS and classification (referring to Table 17). This stage uses the training objective in Equation 7 with a relatively small batch size to alleviate in-batch false negatives, further improving the overall model performance.

**Contrastive Distillation.** Finally, instead of further fine-tuning only with coarse-grained hard signals, the model distills fine-grained soft knowledge from a stronger teacher model, using supervised high-quality data. The student is trained to align its normalized temperature-scaled cosine similarity distribution with that of the teacher. This stage employs the training objectives in Equation 8 and Equation 7 to further improve the model capacity that captures nuanced semantic differences.

The overall workflow of the multi-stage training pipeline is illustrated in Figure 3. The model obtained after pre-training followed by fine-tuning is denoted KaLM-Embedding-V2, and further applying contrastive distillation produces KaLM-Embedding-V2.5.

## 3.4 TRAINING DATA

We curate around 470M samples over 20 categories of large-scale weakly supervised data for pre-training, and about 6M samples over 100 categories of high-quality supervised data for fine-tuning as well as contrastive distillation, with detailed statistics presented in Table 16 and Table 17. Our training datasets cover both retrieval and non-retrieval tasks, including retrieval, classification, clustering, STS, and pair classification. To ensure embeddings with specific task instruction-following abilities, we prepend specific task instructions to the queries. The instructed query is formulated as follows:

$$q_{\text{inst}} = \texttt{Instruct: \{task instruction\} Query: } q. \tag{9}$$

Instructions for different task types are summarized in Table 1, and a detailed task instruction list is provided in Table 18. For symmetric tasks (*e.g.,* STS and Pair Classification), task instructions are also prepended to the passages, whereas for asymmetric tasks, passages remain unchanged.

### 3.4.1 RETRIEVAL DATASETS

We collect diverse and comprehensive retrieval datasets for both pre-training and fine-tuning (see Table 16 and Table 17), and further enrich them via hard negative mining and persona-based synthesis.

**Hard Negative Mining.** As mentioned in §3.2, the training objective is to maximize the similarity between a query and its positive while minimizing similarity to negatives, especially hard negatives. However, most retrieval datasets only provide query-positive pairs. To address this, we mine hard negatives manually. Specifically, a previously trained model is used to retrieve candidate passages, from which we sample 7 negatives ranked between positions 50 and 100.

**Persona-based Synthetic Data.** Following (Wang et al., 2024a), we generate 550k synthetic samples using Qwen2-72B-Instruct, spanning six task types with 40k unique instructions. To further enhance diversity, we incorporate randomly sampled personas from Persona Hub (Chan et al., 2024) as system prompts during instruction generation, thereby enriching domain coverage while avoiding role conflicts in subsequent data generation (Tan et al., 2024).

### 3.4.2 NON-RETRIEVAL DATASETS

In addition to retrieval datasets, we also collect large-scale non-retrieval datasets covering four task types: (1) classification, (2) clustering, (3) semantic textual similarity (STS), and (4) pair classification (see Table 16 and Table 17). To ensure compatibility with contrastive learning, all datasets are reformulated into a unified retrieval-style format: query $q$, positive target $p^+$, and hard negatives $\{p_1^-, p_2^-, \ldots, p_M^-\}$. To accommodate the different formats of these tasks, we process STS and pair classification symmetrically, and clustering/classification asymmetrically, as detailed below.

**Symmetric Data Processing.** To construct training samples for STS and pair classification datasets, we collect any pair of texts with the corresponding relevance score in the range [0, 5], *i.e.*, $(t', t'', score)$, where we create two positive pairs $(q = t', p^+ = t'')$ and $(q = t'', p^+ = t')$ if $score > 4$. Besides, for the dataset with binary labels (0 or 1), we create two positive pairs $(q = t', p^+ = t'')$ and $(q = t'', p^+ = t')$ if $score = 1$. Hard negatives are mined from the candidate pool of other texts using the method proposed in §3.4.1. Task instructions are prepended to both queries, positive targets, as well as hard negatives, because STS and pair classification are symmetric tasks, as shown in Table 1.

**Asymmetric Data Processing.** For clustering and classification datasets, training samples are constructed from text-label pairs $(t, label)$ as $(q = t, p^+ = label)$. Hard negatives are first drawn from other labels within the dataset; if fewer than $M$, additional negatives are sampled from labels across all clustering or classification datasets, mitigating the issue of having too few label categories in certain individual datasets. Task instructions are prepended to queries only in this situation. Inspired by (Lee et al., 2025a), we further apply example-based multi-class labeling: positives are randomly sampled examples from the same cluster/class, while negatives are sampled from other clusters/classes. In this symmetric setting, task instructions are prepended to both the queries, positives, and hard negatives.

## 4 EXPERIMENT

Experimental details, including implementation details, comparison baselines, and evaluation, are provided in Appendix B. The full MTEB results for all tasks, and the statistics of datasets as well as the detailed task instructions, are provided in Appendix H and Appendix I, respectively.

### 4.1 MAIN RESULTS

Table 2 presents the overall comparison of 18 models, reporting the average MTEB scores across all tasks and task types. From the results, we have several key observations: (1) Large-scale open-source models (> 1B parameters) such as Qwen-Embedding-8B, NV-Embed-v2 and bge-multilingual-gemma2 achieve strong results but at a high computational cost. (2) Among models with < 1B parameters, KaLM-Embedding-V2 achieves notable improvements over competitive baselines (*e.g.,* Qwen3-Embedding-0.6B and jina-embeddings-v3), improving over V1 by **+4.37** MTK (cmn) and **+2.53** MTK (eng). (3) KaLM-Embedding-V2.5 further advances SOTA among models with < 1B parameters, with average scores of **70.13** MTK (avg) and **69.16** MTY (avg), competing with billion-scale models while maintaining efficiency. Overall, these results manifest both effectiveness and compactness of the `KaLM-Embedding-V2` series, making it an economical choice for deploying online applications.

Table 3 and Table 4 report detailed task results, where Class., Clust., PairCL., Reran., Retri., STS, and Summ. denote Classification, Clustering, Pair Classification, Reranking, Retrieval, Semantic Textual Similarity, and Summarization. Among models with < 1B parameters, KaLM-Embedding-V2.5 achieves best or second-best results in **6/6** cases on MTEB (cmn, v1) and **4/7** cases on MTEB (eng, v1). Compared to models with > 1B parameters, KaLM-Embedding-V2.5 achieves competitive performance across all tasks on both MTEB (cmn, v1) and MTEB (eng, v1), substantially advancing the development of downstream applications. These results manifest the versatility and compactness of the `KaLM-Embedding-V2` series again. Notably, the `KaLM-Embedding-V2` series is fine-tuned and distilled on just 2-4 GPUs with about 6M samples, compared to Qwen3-Embedding-0.6B's 19M samples, indicating the effectiveness of our superior training techniques and data engineering.

Table 2: Evaluation results on MTEB Chinese (cmn) and English (eng). The best results are **boldfaced** and the second-best ones are underlined (only considering models with < 1B parameters). The `KaLM-Embedding-V2` series achieves SOTA performance among competitive embedding models with <1B parameters, serving as an economical choice for building online applications, *e.g.,* RAG systems. 'M' and 'B' denote million and billion, respectively. MTK refers to Mean (Task), MTY to Mean (Type). Results are mainly sourced from MTEB leaderboard (accessed Sep 10, 2025).

| Model | Size | Dim | MTEB (cmn, v1) | | MTEB (eng, v1) | | Avg | |
|---|---|---|---|---|---|---|---|---|
| | | | MTK | MTY | MTK | MTY | MTK | MTY |
| **Commercial embedding API services** | | | | | | | | |
| text-embedding-3-large (2024) | - | 3072 | - | - | 64.52 | 62.33 | - | - |
| Cohere-embed-multilingual-v3.0 (2023) | - | 1024 | - | - | 64.01 | 62.09 | - | - |
| **Open-Source Embedding Models > 1B parameters** | | | | | | | | |
| GRITLM 8X7B (13B active) (2024) | 13B | 4096 | - | - | 65.50 | 63.01 | - | - |
| bge-multilingual-gemma2 (2024) | 9B | 3584 | 67.64 | 68.52 | 69.88 | 66.11 | 68.76 | 67.32 |
| NV-Embed-v2 (2025a) | 7B | 4096 | - | - | 72.31 | 67.97 | - | - |
| Qwen3-Embedding-8B (2025c) | 8B | 4096 | 73.84 | 75.00 | - | - | - | - |
| e5-mistral-7b-instruct (2022) | 7B | 4096 | 59.92 | 60.51 | 66.46 | 64.22 | 63.19 | 62.37 |
| Qwen3-Embedding-4B (2025c) | 4B | 2560 | 72.26 | 73.50 | - | - | - | - |
| gte-Qwen-1.5B-instruct (2023) | 1.5B | 1536 | 67.12 | 67.83 | 67.19 | 64.44 | 67.16 | 66.14 |
| **Open-Source Embedding Models < 1B parameters** | | | | | | | | |
| Qwen3-Embedding-0.6B (2025c) | 596M | 1024 | 66.33 | 67.44 | 66.76 | 63.62 | 66.55 | 65.53 |
| jina-embeddings-v3 (Multi-LoRA) (2024) | 572M | 1024 | 61.82 | 61.61 | 65.51 | 62.76 | 63.67 | 62.19 |
| multilingual-e5-large (2024b) | 560M | 1024 | 58.08 | 58.24 | 60.89 | 59.48 | 59.49 | 58.86 |
| bge-m3 (Dense) (2024) | 560M | 1024 | 60.34 | 61.23 | 59.84 | 58.98 | 60.09 | 60.11 |
| paraphrase-ML-mpnet-base-v2 (2019) | 278M | 768 | 42.89 | 48.36 | 54.64 | 55.46 | 48.77 | 51.91 |
| gte-multilingual-base (Dense) (2024) | 305M | 768 | 62.94 | 63.92 | 61.40 | 60.10 | 62.17 | 62.01 |
| **KaLM Embedding series** | | | | | | | | |
| KaLM-Embedding-V1 | 494M | 896 | 63.78 | 64.56 | 64.94 | 61.49 | 64.36 | 63.03 |
| KaLM-Embedding-V2 | 494M | 896 | 68.15 | 69.28 | 67.47 | 64.14 | 67.81 | 66.71 |
| KaLM-Embedding-V2.5 | 494M | 896 | **70.93** | **72.46** | **69.33** | **65.83** | **70.13** | **69.16** |

Table 3: Detailed model performance on MTEB (cmn, v1) derived from C-MTEB (Xiao et al., 2024).

| Model | Size | MTEB (cmn, v1) | | | | | | | |
|---|---|---|---|---|---|---|---|---|---|
| | | MTK | MTY | Class. | Clust. | PairCl. | Reran. | Retri. | STS |
| bge-multilingual-gemma2 | 9B | 67.64 | 68.52 | 75.31 | 59.30 | 79.30 | 68.28 | 73.73 | 55.19 |
| Qwen3-Embedding-8B | 8B | 73.84 | 75.00 | 76.97 | 80.08 | 84.23 | 66.99 | 78.21 | 63.53 |
| e5-mistral-7b-instruct | 7B | 59.92 | 60.51 | 72.96 | 52.30 | 66.31 | 61.38 | 61.75 | 48.34 |
| Qwen3-Embedding-4B | 4B | 72.26 | 73.50 | 75.46 | 77.89 | 83.34 | 66.05 | 77.03 | 61.26 |
| gte-Qwen2-1.5B-instruct | 1.5B | 67.12 | 67.83 | 72.53 | 54.61 | 79.50 | 68.21 | 71.86 | 60.25 |
| Qwen3-Embedding-0.6B | 596M | 66.33 | 67.44 | 71.40 | 68.74 | 76.42 | 62.58 | 71.03 | 54.52 |
| jina-embeddings-v3 (Multi-LoRA) | 572M | 61.82 | 61.61 | 70.47 | 50.22 | 67.22 | 60.72 | 68.54 | 52.46 |
| multilingual-e5-large | 560M | 58.08 | 58.24 | 69.80 | 48.23 | 64.52 | 57.45 | 63.65 | 45.81 |
| bge-m3 (Dense) | 560M | 60.34 | 61.23 | 70.52 | 45.75 | 73.98 | 62.88 | 65.43 | 48.79 |
| paraphrase-ML-mpnet-base-v2 | 278M | 42.89 | 48.36 | 65.88 | 39.67 | 80.90 | 44.91 | 22.92 | 35.85 |
| gte-multilingual-base (Dense) | 305M | 62.94 | 63.92 | 66.84 | 47.48 | 78.34 | **68.17** | 71.95 | 50.75 |
| KaLM-Embedding-V1 | 494M | 63.78 | 64.56 | 73.89 | 57.54 | 72.94 | 64.48 | 70.12 | 48.41 |
| KaLM-Embedding-V2 | 494M | 68.15 | 69.28 | 75.14 | 69.76 | 77.91 | 65.16 | 72.15 | 55.58 |
| KaLM-Embedding-V2.5 | 494M | **70.93** | **72.46** | **77.48** | **73.09** | **84.09** | 66.90 | **73.42** | **59.80** |

## 4.2 IN-DEPTH ANALYSIS

We next investigate how different key settings influence model performance, including (1) focal-style reweighting, (2) online hard negative mixing, (3) bidirectional attention, (4) example-based multi-class labeling, (5) contrastive distillation, and (6) the temperature coefficient.

**Ablation Study on Training Techniques.** Table 5 and Figure 4 present the ablation results on both MTEB (eng, v1) and MTEB (cmn, v1). We observe that removing focal-style reweighting leads to the largest performance drop, with MTK dropping from 69.33 to 68.70 on eng and from

Table 4: Detailed embedding model performance on MTEB (eng, v1) (Muennighoff et al., 2023a). Results on MTEB (eng, v2) (Enevoldsen et al., 2025) are provided in Table 12.

| Model | Size | MTEB (eng, v1) | | | | | | | | |
|---|---|---|---|---|---|---|---|---|---|---|
| | | MTK | MTY | Class. | Clust. | PairCl. | Reran. | Retri. | STS | Summ. |
| text-embedding-3-large | - | 64.52 | 62.33 | 75.12 | 49.01 | 85.81 | 59.16 | 55.43 | 81.73 | 30.05 |
| Cohere-embed-multilingual-v3.0 | - | 64.01 | 62.09 | 76.01 | 46.60 | 86.15 | 57.86 | 53.84 | 83.15 | 30.99 |
| GRITLM 8X7B | 13B | 65.50 | 63.01 | 77.69 | 50.14 | 85.23 | 59.80 | 55.13 | 83.26 | 29.82 |
| bge-multilingual-gemma2 | 9B | 69.88 | 66.11 | 88.08 | 54.65 | 85.97 | 59.72 | 59.24 | 83.88 | 31.20 |
| NV-Embed-v2 | 7B | 72.31 | 67.97 | 90.37 | 58.46 | 88.67 | 60.65 | 62.65 | 84.31 | 30.7 |
| e5-mistral-7b-instruct | 7B | 66.46 | 64.22 | 77.37 | 50.26 | 88.42 | 60.21 | 57.07 | 84.65 | 31.53 |
| gte-Qwen2-1.5B-instruct | 1.5B | 67.19 | 64.44 | 82.53 | 48.75 | 87.52 | 59.98 | 58.29 | 82.81 | 31.17 |
| Qwen3-Embedding-0.6B | 596M | 66.76 | 63.62 | 82.61 | 49.87 | 84.29 | 57.96 | 54.32 | 86.97 | 29.23 |
| jina-embeddings-v3 (Multi-LoRA) | 572M | 65.51 | 62.76 | 82.58 | 45.21 | 84.01 | 58.13 | 53.88 | 85.81 | 29.71 |
| multilingual-e5-large | 560M | 60.89 | 59.48 | 71.77 | 41.23 | 84.75 | 55.96 | 51.40 | 81.62 | 29.64 |
| bge-m3 (Dense) | 560M | 59.84 | 58.98 | 74.08 | 37.27 | 84.50 | 55.28 | 48.82 | 81.37 | 31.55 |
| paraphrase-ML-mpnet-base-v2 | 278M | 54.64 | 55.46 | 67.46 | 38.50 | 80.81 | 53.80 | 35.34 | 80.77 | 31.57 |
| gte-multilingual-base (Dense) | 305M | 61.40 | 60.10 | 70.89 | 44.31 | 84.23 | 57.47 | 51.08 | 82.11 | 30.58 |
| KaLM-Embedding-V1 | 494M | 64.94 | 61.49 | 84.74 | 47.82 | 83.26 | 55.41 | 51.65 | 82.24 | 25.23 |
| KaLM-Embedding-V2 | 494M | 67.47 | 64.14 | 87.19 | 56.05 | 86.18 | 56.74 | 51.67 | 82.61 | 28.51 |
| KaLM-Embedding-V2.5 | 494M | 69.33 | 65.83 | 88.34 | 56.59 | 86.60 | 57.84 | 55.00 | 85.27 | 31.18 |

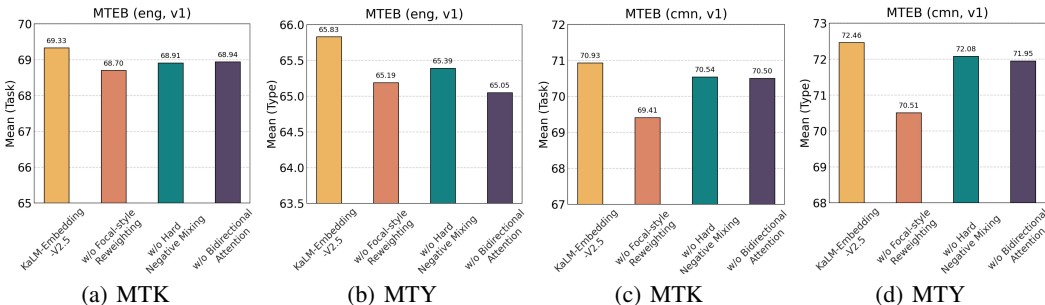

| (a) MTK | (b) MTY | (c) MTK | (d) MTY |
|---|---|---|---|

Figure 4: Ablation study on focal-style reweighting, hard negative mixing, and bidirectional attention.

70.93 to 69.41 on cmn, indicating that it plays a key role in improving general performance. On the other hand, eliminating hard negative mixing or bidirectional attention yields smaller but consistent declines, demonstrating that hard negative mixing supplements informative hard negatives throughout training, while embeddings generated with bidirectional attention are more effective than those generated with causal attention. Overall, these results confirm that the proposed training techniques are complementary and jointly contribute to the performance of the `KaLM-Embedding-V2` series.

**Example-based *v.s.* Label-based Labeling.** Table 6 presents the comparison results between using class/clust and sampled examples as positives and negatives. Note that, in the setting of 'Example', both example-based and label-based labeling data are used for training. The results demonstrate that example-based labeling leads to considerable improvements, especially on the clustering task, demonstrating the effect of supplementing the class. and clust. data with example-based multi-class labeling.

Table 6: Effect of example-based labeling.

| Setting | MTEB (cmn, v1) | | MTEB (eng, v1) | |
|---|---|---|---|---|
| | Class. | Clust. | Class. | Clust. |
| Example | 77.48 | 73.09 | 88.34 | 56.59 |
| Label | 76.90 | 64.71 | 87.03 | 52.71 |

**Effectiveness of Contrastive Distillation.** During the contrastive distillation stage, the KaLM-Embedding-V2 is further optimized using the training objectives in Equation 8 (denoted as 'KL') and Equation 7 (denoted as 'CL'). Implementation details can be seen in Appendix B. To assess the contribution of each objective, we conduct an ab-

Table 7: Effect of contrastive distillation.

| Setting | MTEB (cmn, v1) | | MTEB (eng, v1) | |
|---|---|---|---|---|
| | MTK | MTY | MTK | MTY |
| CL+KL | 70.93 | 72.46 | 69.33 | 65.83 |
| only KL | 70.72 | 72.48 | 68.63 | 65.29 |
| only CL | 68.31 | 69.88 | 67.67 | 64.37 |

Table 5: Detailed ablation study results on several key components, including focal-style reweighting, hard negative mixing, and bidirectional attention.

| | MTEB (eng, v1) | | | | | | | | |
|---|---|---|---|---|---|---|---|---|---|
| Row Setting | MTK | MTY | Class. | Clust. | PairCl. | Reran. | Retri. | STS | Summ. |
| 1 KaLM-Embedding-V2.5 | 69.33 | 65.83 | 88.34 | 56.59 | 86.60 | 57.84 | 55.00 | 85.27 | 31.18 |
| 2   w/o Focal-style Reweighting | 68.70 | 65.19 | 87.68 | 55.40 | 86.62 | 57.66 | 54.82 | 84.31 | 29.86 |
| 3   w/o Hard Negative Mixing | 68.91 | 65.39 | 87.88 | 55.81 | 86.67 | 57.46 | 54.91 | 84.64 | 30.38 |
| 4   w/o Bidirectional Attention | 68.94 | 65.05 | 88.51 | 56.10 | 85.40 | 57.65 | 54.70 | 84.55 | 28.43 |
| | MTEB (cmn, v1) | | | | | | | | |
| 1 KaLM-Embedding-V2.5 | 70.93 | 72.46 | 77.48 | 73.09 | 84.09 | 66.90 | 73.42 | 59.80 | - |
| 2   w/o Focal-style Reweighting | 69.41 | 70.51 | 76.31 | 70.07 | 79.66 | 65.58 | 71.73 | 59.71 | - |
| 3   w/o Hard Negative Mixing | 70.54 | 72.08 | 76.71 | 72.02 | 84.28 | 66.50 | 73.26 | 59.70 | - |
| 4   w/o Bidirectional Attention | 70.50 | 71.95 | 77.41 | 72.71 | 82.87 | 66.40 | 73.01 | 59.27 | - |

lation study, as shown in Table 7. The results show that combining CL and KL achieves the best performance. Using only CL leads to the largest drop, while using only KL yields smaller but consistent declines, especially in MTEB (eng, v1). This means that KL serves as the primary learning signal, while CL provides the auxiliary learning one, and their combination yields the best performance.

**Sensitivity of Temperature Coefficient.** KL-divergence is sensitive to the temperature coefficient (coef) (Hinton et al., 2015). Table 8 shows the performance in terms of different $\tau$ under the 'only KL' setting, where $\tau = 0.01$ (Low), $\tau = 0.05$ (Mid), and $\tau = 0.1$ (High). We can observe that Mid leads to the best performance,

Table 8: Sensitivity of temperature coef $\tau$.

| Setting | MTEB (cmn, v1) | | MTEB (eng, v1) | |
|---|---|---|---|---|
| | MTK | MTY | MTK | MTY |
| Low | 68.06 | 69.54 | 67.85 | 64.80 |
| Mid | 70.72 | 72.48 | 68.63 | 65.29 |
| High | 67.10 | 68.28 | 66.60 | 63.72 |

since setting $\tau$ to a too small value (*e.g.*, 0.01) makes the teacher distribution overly skewed, while a too large $\tau$ (such as 0.1) oversmooths it, both reducing the informativeness of the learning signals.

## 5  CONCLUSION

In this work, we propose `KaLM-Embedding-V2`, a series of versatile and compact embedding models that achieve SOTA performance on MTEB (cmn, v1) and MTEB (eng, v1) among competitive embedding models with < 1B parameters. The strong performance stems from several systematized innovative designs. For model architecture, we remove the causal attention mask to enable more effective representation learning. For training techniques, we introduce a multi-stage training pipeline that progressively incentivizes advanced embedding capabilities in LLMs. For training objectives, we introduce a focal-style reweighting mechanism to emphasize difficult samples, and an online hard-negative mixing strategy to enrich hard negatives. For training data, we collect over 20 categories of data for pre-training and 100 categories of data for fine-tuning as well as distillation, leveraging task-specific instructions, hard-negative mining, example-based multi-class labeling, etc, to carefully curate data. By combining superior training techniques and high-quality data, `KaLM-Embedding-V2` significantly outperforms others of comparable size and even competes with 3x to 26x larger models.

## ACKNOWLEDGMENTS

This work was jointly supported by grants: Natural Science Foundation of China (No. 62422603, No. 62376067) and Guangdong Basic and Applied Basic Research Foundation (No. 2024B0101050003).

## REPRODUCIBILITY STATEMENT

To ensure the reproducibility of our work and to facilitate a clearer understanding of our contributions, we provide extensive supporting materials. In the main text, we describe our proposed method in §3 and present the detailed benchmark results in §4. In the Appendix B and Appendix I, we provide further detailed information, including implementation details, training details, and evaluation details, statistics of datasets, and task instructions used in evaluations, to ensure our results are reproducible.

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

## A    THE USE OF LARGE LANGUAGE MODELS

In this work, we utilized LLMs solely for the purpose of polishing writing. The LLMs were not used for content generation, and all research, analysis, and conclusions presented are the result of our own work and independent thought.

## B    EXPERIMENTAL DETAILS

**Implementation Details.** We adopt InfoNCE loss (Gutmann & Hyvärinen, 2010) and KL-divergence loss (Hinton et al., 2015) as training objectives, with temperature coefficients $\tau$ set to 0.01 and 0.05, respectively. Qwen2-0.5 (Yang et al., 2024) serves as the base decoder-only LLM backbone, combined with a simple yet effective mean pooling. To enable fully bidirectional modeling, we remove the causal attention mask from the decoder-only LLM. The embedding dimension is 896, with a maximum input length of 512 tokens. The model is fully fine-tuned with all parameters updated, using mixed precision with Bfloat16. Matryoshka Representation Learning (MRL) (Kusupati et al., 2022) is applied to both InfoNCE and KL-divergence losses with embedding dimensions of 896, 512, 256, 128, and 64, weighted by 1.0, 0.3, 0.2, 0.1, and 0.1, respectively. The model is optimized by the Adam optimizer (Kingma & Ba, 2015).

Based on the above common configurations, we detail the settings for each training stage. **(1) Pre-training:** We exclusively use in-batch negatives for training efficiency. Pre-training is conducted on 6 nodes (8 GPUs each) for 1 epoch, corresponding to approximately 19k steps, with a per-GPU batch size of 512 and a learning rate of 1e-4. **(2) Fine-tuning:** We incorporate hard negatives by sampling $M = 7$ examples from ranks 50 to 100 within the candidate pool. Training is conducted for 1 epoch, approximately 12k steps, with a per-GPU batch size of 120 and a learning rate of 2e-5. The focusing parameter $\gamma$ in Equation 4 is set to 0.5. For each sample, a pair-wise and a list-wise hard negative is mined. Fine-tuning is performed on 4 GPUs, requiring approximately 220 GPU hours for 1 epoch. **(3) Contrastive distillation:** The model is jointly optimized with contrastive and KL-divergence losses, weighted at 0.3 and 0.7, respectively. Qwen3-Embedding-8B (Zhang et al., 2025c) is used as the teacher model, where teacher embeddings for all training samples are pre-computed and cached to accelerate training. Training is run for 1 epoch, approximately 24k steps, with a per-GPU batch size of 120 and a learning rate of 1e-5. Distillation is performed on just 2 GPUs, requiring about 280 GPU hours for 1 epoch. The detailed hyperparameter settings adopted in the experiments are presented in Table 9.

Table 9: Hyperparameters used in the experiments. For batch size, training steps, learning rate, and so on, the three values correspond to pre-training, fine-tuning, and contrastive distillation, respectively.

| Parameter | Value |
|---|---|
| Batch size (per GPU) | 512/120/120 |
| GPU used | 48/4/2 |
| Training Steps | 19k/12k/24k |
| Training Data Size | 470M/6M/6M |
| Warm-up steps | 10% / 200 / 200 |
| Learning Rate | 1e-4/2e-5/1e-5 |
| Epochs | 1 (all stages) |
| Base model | Qwen2-0.5 (bidirectional) |
| Pooling strategy | Mean pooling |
| Embedding dimension | 896 |
| Maximum Input Length | 512 |
| MRL Dimensions | 896, 512, 256, 128, 64 |
| MRL Weights | 1.0, 0.3, 0.2, 0.1, 0.1 |
| Focusing Parameter $\gamma$ | 0.5 |
| Hard negatives | $M = 7$, ranks 50-100 |
| Optimizer | Adam |
| Precision | Bfloat16 |
| Temperature Coefficients | Contrastive Learning - 0.01 Contrastive Distillation - 0.05 |
| Teacher model | Qwen3-Embedding-8B |

**Baselines.** We compare the `KaLM-Embedding-V2` series with the following competitive general-purpose and multilingual open-source text embedding models and commercial embedding API services. The open-source models include: paraphrase-multilingual (ML)-mpnet-base-v2 (Reimers & Gurevych, 2019), jina-embeddings-v3 (Sturua et al., 2024), Qwen3-Embedding-8B/Qwen3-Embedding-4B/Qwen3-Embedding-0.6B/gte-multilingual-base/gte-Qwen2-7B-instruct/gte-Qwen2-1.5B-instruct (Zhang et al., 2025c; Li et al., 2023; Zhang et al., 2024), bge-m3/bge-multilingual-gemma2/bge-large-en-v1.5 (Chen et al., 2024; Xiao et al., 2024), EmbeddingGemma-300M (Vera et al., 2025), multilingual-e5-large(-instruct)/e5-mistral-7b-instruct (Wang et al., 2024b; 2022), GRITLM 8X7B (Muennighoff et al., 2024) (a sparse mixture-of-experts embedding model with 13B active parameters during inference), NV-Embed-v2 (Lee et al., 2025a), and KaLM-Embedding-V1 (Hu et al., 2025). The commercial embedding services include text-embedding-3-large (OpenAI, 2024) from OpenAI and Cohere-embed-multilingual-v3.0 (Cohere, 2023).

**Evaluation.** We evaluate the `KaLM-Embedding-V2` series and the competitive baseline embedding models on MTEB (Muennighoff et al., 2023a; Xiao et al., 2024) for both Chinese (cmn) and English (eng). For Chinese, we use MTEB (cmn v1), derived from C-MTEB (Xiao et al., 2024), which comprises 35 tasks across 6 task types. For English, we adopt MTEB (eng v1) (Muennighoff et al., 2023a), covering 56 tasks across 7 task types, providing a broader evaluation scope than v2, which contains only 41 tasks across the same number of task types. Following the MTEB (cmn, v1) leaderboard, we exclude AmazonReviewsClassification, MassiveIntentClassification, and MassiveScenarioClassification from the classification task, as well as STS22 from the STS task, resulting in 31 tasks. This setup slightly differs from the original C-MTEB (Xiao et al., 2024). For evaluation, we evaluate our `KaLM-Embedding-V2` series using a maximum length of 512 tokens to ensure fair comparison with previous works. For models without officially reported results on the MTEB leaderboards, we evaluate them using the task instructions summarized in Table 18 to ensure fair comparison. We also provide results on MTEB (eng, v2) (Enevoldsen et al., 2025) in Table 12.

Table 10: OOD Evaluation on real-world industrial scenarios. Recall@K measures whether the positive item appears in the top-K retrieved items. MRR@K denotes mean reciprocal rank and further measures the ranking quality. It reciprocally discounts the position.

| Model | Size | MRR@1 | MRR@5 | MRR@10 | Recall@1 | Recall@5 | Recall@10 |
|---|---|---|---|---|---|---|---|
| **Customer Service FAQ Retrieval** | | | | | | | |
| Qwen3-Embedding-8B | 7.57B | 44.49 | **57.79** | **58.91** | 44.49 | **78.44** | **86.69** |
| Qwen3-Embedding-0.6B | 596M | 40.36 | 53.60 | 54.61 | 40.36 | 75.22 | 82.56 |
| bge-m3 (Dense) | 560M | 34.40 | 46.68 | 48.19 | 34.40 | 68.80 | 79.81 |
| gte-multilingual-base (Dense) | 305M | 39.90 | 50.44 | 51.47 | 39.90 | 67.43 | 75.68 |
| KaLM-Embedding-V2.5 | 494M | **45.87** | 56.96 | 58.05 | **45.87** | 77.06 | 85.32 |
| **Game Documentation Search** | | | | | | | |
| Qwen3-Embedding-8B | 7.57B | 23.61 | 35.64 | 37.52 | 23.61 | 56.55 | 70.45 |
| Qwen3-Embedding-0.6B | 596M | 20.70 | 31.40 | 33.14 | 20.70 | 50.23 | 63.28 |
| bge-m3 (Dense) | 560M | 20.02 | 30.62 | 32.47 | 20.02 | 49.04 | 62.70 |
| gte-multilingual-base (Dense) | 305M | 18.10 | 27.50 | 29.02 | 18.10 | 43.86 | 55.14 |
| KaLM-Embedding-V2.5 | 494M | **23.82** | **36.36** | **38.24** | **23.82** | **58.23** | **72.22** |

## C OUT-OF-DOMAIN GENERATION

To comprehensively assess robustness and generalization in real-world industrial applications, we conducted out-of-domain (OOD) evaluations in two Chinese retrieval scenarios, with sizes ranging from thousands to tens of thousands. The first involves customer service FAQ retrieval, where all queries originate from real user interactions, with relevance labels manually annotated by human experts. The second targets game documentation search in a vertical domain, utilizing real user-generated queries; relevant documents were filtered and selected based on user click-through data. None of the models has been trained on these datasets, ensuring genuine OOD evaluation. We choose embedding models widely used in industries from GTE and BGE as baselines. From the results shown in Table 10, KaLM-Embedding-V2.5 achieves SOTA performance compared to models of comparable size. Furthermore, despite being 15 times smaller in size than Qwen3-Embedding-

Table 11: Matryoshka embedding performance, where 'Full' denotes the maximum dimension, specifically 896 for the KaLM-Embedding series.

| Model | Dim | MTK | MTY | Class. | Clust. | PairCl. | Reran. | Retri. | STS | Summ. |
|---|---|---|---|---|---|---|---|---|---|---|
| **MTEB (eng, v1)** | | | | | | | | | | |
| | Full | 69.33 | 65.83 | 88.34 | 56.59 | 86.60 | 57.84 | 55.00 | 85.27 | 31.18 |
| | 512 | 69.13 (-0.288%) | 65.65 | 88.35 | 56.52 | 86.53 | 57.76 | 54.44 | 85.32 | 30.65 |
| KaLM-Embedding-V2.5 | 256 | 68.80 (-0.764%) | 65.43 | 88.29 | 56.37 | 86.35 | 57.45 | 53.47 | 85.12 | 30.95 |
| | 128 | 68.05 (-1.846%) | 64.95 | 88.14 | 56.29 | 85.83 | 56.64 | 51.25 | 84.95 | 31.57 |
| | 64 | 66.44 (-4.168%) | 63.63 | 87.87 | 56.06 | 84.96 | 56.04 | 46.63 | 84.13 | 29.71 |
| | Full | 69.36 | 65.86 | 88.55 | 56.18 | 86.86 | 57.86 | 55.14 | 85.36 | 31.07 |
| KaLM-Embedding-V2.5 | 512 | 69.02 (-0.490%) | 65.71 | 88.71 | 56.66 | 86.92 | 58.13 | 53.67 | 84.77 | 31.11 |
| (w/o MKL) | 256 | 68.40 (-1.384%) | 65.34 | 88.69 | 56.63 | 86.58 | 57.64 | 52.10 | 83.93 | 31.80 |
| | 128 | 67.36 (-2.884%) | 64.40 | 88.61 | 56.45 | 85.59 | 56.61 | 49.40 | 83.29 | 30.84 |
| | 64 | 65.36 (-5.767%) | 63.01 | 88.36 | 56.00 | 84.39 | 55.84 | 43.76 | 82.05 | 30.68 |
| | Full | 67.47 | 64.14 | 87.19 | 56.05 | 86.18 | 56.74 | 51.67 | 82.61 | 28.51 |
| | 512 | 67.23 (-0.356%) | 63.98 | 87.14 | 56.04 | 86.11 | 56.49 | 50.90 | 82.62 | 28.57 |
| KaLM-Embedding-V2 | 256 | 66.76 (-1.052%) | 63.76 | 87.18 | 56.03 | 85.83 | 56.09 | 49.55 | 82.19 | 29.43 |
| | 128 | 65.65 (-2.697%) | 62.83 | 86.98 | 55.80 | 84.94 | 55.09 | 46.39 | 81.92 | 28.67 |
| | 64 | 63.73 (-5.543%) | 61.56 | 86.72 | 55.53 | 83.63 | 54.21 | 40.83 | 80.79 | 29.19 |
| | Full | 64.94 | 61.49 | 84.74 | 47.82 | 83.26 | 55.41 | 51.65 | 82.24 | 25.23 |
| | 512 | 64.48 (-0.708%) | 61.14 | 84.60 | 47.49 | 82.92 | 54.72 | 50.74 | 81.90 | 25.61 |
| KaLM-Embedding-V1 | 256 | 63.85 (-1.678%) | 60.85 | 84.29 | 47.21 | 82.74 | 53.94 | 49.01 | 81.90 | 26.89 |
| | 128 | 62.13 (-4.327%) | 59.35 | 83.71 | 46.44 | 81.09 | 52.05 | 44.83 | 81.40 | 25.96 |
| | 64 | 59.69 (-8.084%) | 57.71 | 82.68 | 45.49 | 78.54 | 50.41 | 38.61 | 80.60 | 27.64 |
| **MTEB (cmn, v1)** | | | | | | | | | | |
| | Full | 70.93 | 72.46 | 77.48 | 73.09 | 84.09 | 66.90 | 73.42 | 59.80 | - |
| | 512 | 70.80 (-0.183%) | 72.36 | 77.48 | 73.07 | 84.05 | 66.83 | 72.96 | 59.79 | - |
| KaLM-Embedding-V2.5 | 256 | 70.43 (-0.705%) | 72.09 | 77.38 | 73.06 | 84.21 | 66.20 | 71.94 | 59.73 | - |
| | 128 | 69.76 (-1.650%) | 71.62 | 77.38 | 73.37 | 84.05 | 65.68 | 69.60 | 59.61 | - |
| | 64 | 68.10 (-3.990%) | 70.32 | 76.98 | 73.17 | 83.95 | 63.60 | 65.06 | 59.13 | - |
| | Full | 70.91 | 72.46 | 77.44 | 72.80 | 84.53 | 66.74 | 73.45 | 59.79 | - |
| KaLM-Embedding-V2.5 | 512 | 70.45 (-0.649%) | 71.84 | 77.73 | 72.26 | 82.38 | 66.59 | 72.96 | 59.12 | - |
| (w/o MKL) | 256 | 69.89 (-1.438%) | 71.38 | 77.67 | 72.25 | 82.21 | 65.80 | 71.65 | 58.67 | - |
| | 128 | 68.75 (-3.046%) | 70.36 | 77.50 | 72.03 | 81.25 | 64.30 | 68.98 | 58.08 | - |
| | 64 | 66.89 (-5.669%) | 68.91 | 77.17 | 71.83 | 80.48 | 63.01 | 63.80 | 57.14 | - |
| | Full | 68.15 | 69.28 | 75.14 | 69.76 | 77.91 | 65.16 | 72.15 | 55.58 | - |
| | 512 | 67.85 (-0.440%) | 69.01 | 75.04 | 69.35 | 77.64 | 65.09 | 71.46 | 55.50 | - |
| KaLM-Embedding-V2 | 256 | 67.37 (-1.145%) | 68.64 | 74.96 | 69.32 | 77.77 | 64.80 | 69.65 | 55.31 | - |
| | 128 | 66.38 (-2.597%) | 67.88 | 74.85 | 69.41 | 76.93 | 64.15 | 66.92 | 55.02 | - |
| | 64 | 64.13 (-5.899%) | 66.14 | 74.62 | 69.35 | 76.33 | 61.99 | 60.43 | 54.12 | - |
| | Full | 63.78 | 64.56 | 73.89 | 57.54 | 72.94 | 64.48 | 70.12 | 48.41 | - |
| | 512 | 63.39 (-0.611%) | 64.18 | 73.58 | 57.26 | 72.54 | 63.98 | 69.39 | 48.35 | - |
| KaLM-Embedding-V1 | 256 | 62.82 (-1.505%) | 63.77 | 73.71 | 57.20 | 72.56 | 63.50 | 67.50 | 48.17 | - |
| | 128 | 61.59 (-3.434%) | 62.75 | 73.51 | 57.52 | 71.62 | 62.08 | 63.97 | 47.82 | - |
| | 64 | 58.98 (-7.526%) | 60.74 | 72.85 | 56.58 | 71.27 | 60.22 | 56.72 | 46.82 | - |

8B, KaLM-Embedding-V2.5 still outperforms it in 8/12 cases. These results demonstrate that our `KaLM-Embedding-V2` models not only achieve state-of-the-art performance on MTEB, but also exhibit strong generalization and robustness in real-world industrial applications.

# D  MATRYOSHKA EMBEDDING

To enable flexible-dimensional embeddings, we incorporate MRL into both contrastive and KL loss. Unlike previous works, we also optimize matryoshka embeddings using the matryoshka KL objective, referred to as MKL[1]. To verify the effectiveness of matryoshka embeddings and MKL,

---

[1] Note that the teacher model still computes soft signals using the full dimension.

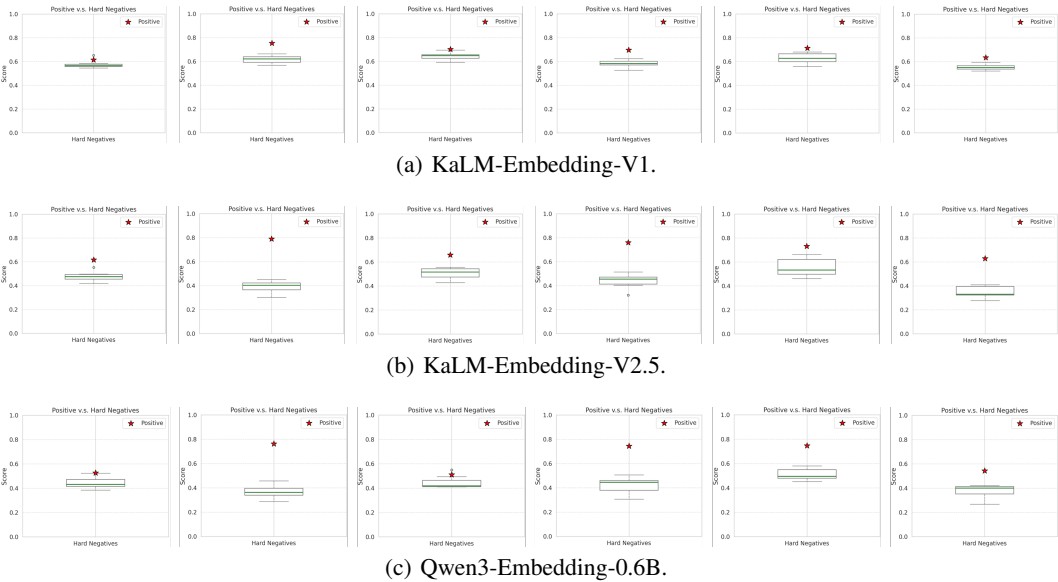

(a) KaLM-Embedding-V1.

(b) KaLM-Embedding-V2.5.

(c) Qwen3-Embedding-0.6B.

Figure 5: Comparison of discriminative capacity between positive and hard negatives. Cases are randomly sampled from the HotpotQA dataset, where the task instruction is "Instruct: Given a query, retrieve documents that answer the query. \n Query: {query }".

we conduct dimensionality reduction experiments along with MKL ablation studies, as shown in Table 11. From the results, we mainly have the following observations. Firstly, for tasks such as Class., Clust., PairCl., STS, and Summ., performance degrades only slightly when using matryoshka embeddings of smaller sizes, whereas tasks like Reran. and Retri. exhibit more substantial drops. This indicates that semantic matching tasks (*e.g.,* Class., Clust., and PairCl.) can be effectively handled even with low-dimensional matryoshka embeddings, whereas retrieval and reranking tasks demand higher-dimensional embeddings to preserve performance. Secondly, compared with KaLM-Embedding-V2.5 (w/o MKL), V2, and V1, KaLM-Embedding-V2.5 demonstrates consistently smaller performance degradation as embedding dimensionality decreases. For example, on MTEB (cmn, v1), the performance drop from full dimension to 64 dimensions is only -3.99% for KaLM-Embedding-V2.5, compared to -5.67% for its counterpart without MKL. We find that the superior robustness of KaLM-Embedding-V2.5 using matryoshka embeddings of smaller sizes mainly stems from its relatively smaller performance degradation on Reran. and Retri. tasks compared to others. These results show that MKL makes KaLM-Embedding-V2.5 more robust, with smaller drops under small embedding dimensions. Thirdly, retrieval tasks exhibit the largest performance drops as embedding dimensions decrease, showing they rely heavily on high-dimensional embedding. This also explains why small, low-dimensional embedding models lag behind larger, high-dimensional ones on retrieval tasks, as illustrated in Table 4. Overall, these results indicate that matryoshka embeddings provide flexible, compact representations that maintain strong performance on semantic matching tasks, while retrieval and reranking tasks benefit from higher-dimensional embeddings.

## E  CASE STUDY

To provide a more intuitive and qualitative understanding of our model's discriminative capacity, we conduct a case study on randomly sampled examples from the HotpotQA, a representative retrieval dataset. For each case, we compute similarity scores between a query, its ground-truth positive, and 7 hard negatives. To visualize the results, the score between the query and the positive is plotted as a single point, *i.e.,* the red star. The seven scores between the query and the hard negatives are used to generate a box plot. An ideal embedding model should assign a significantly higher score to the positive compared to all hard negatives, placing the red star well above the corresponding box plot. This visualization provides a clear comparison of how effectively each model can distinguish the positive passages from hard negative ones. From the results shown in Figure 5,

Table 12: Detailed embedding model performance on MTEB (eng, v2) (Enevoldsen et al., 2025).

| Model | Size | MTEB (eng, v2) | | | | | | | |
|---|---|---|---|---|---|---|---|---|---|
| | | MTK | MTY | Class. | Clust. | PairCl. | Reran. | Retri. | STS | Summ. |
| Qwen3-Embedding-8B | 8B | 75.22 | 68.70 | 90.43 | 58.57 | 87.52 | 51.56 | 69.44 | 88.58 | 34.83 |
| NV-Embed-v2 | 7B | 69.81 | 65.00 | 87.19 | 47.66 | 88.69 | 49.61 | 62.84 | 83.82 | 35.21 |
| Qwen3-Embedding-4B | 4B | 74.60 | 68.09 | 89.84 | 57.51 | 87.01 | 50.76 | 68.46 | 88.72 | 34.39 |
| gte-Qwen2-1.5B-instruct | 1.5B | 67.20 | 63.26 | 85.84 | 53.54 | 87.52 | 49.25 | 50.25 | 82.51 | 33.94 |
| Qwen3-Embedding-0.6B | 596M | 70.70 | 64.88 | 85.76 | 54.05 | 84.37 | 48.18 | **61.83** | **86.57** | 33.43 |
| multilingual-e5-large-instruct | 560M | 65.53 | 61.21 | 75.54 | 49.89 | 86.24 | **48.74** | 53.47 | 84.72 | 29.89 |
| bge-large-en-v1.5 | 335M | 65.89 | 61.87 | 78.34 | 48.01 | 87.13 | 48.26 | 55.44 | 82.79 | 33.13 |
| EmbeddingGemma-300M | 307M | 69.67 | 65.11 | 87.55 | 56.55 | 87.29 | 47.43 | 55.69 | 83.61 | **37.64** |
| KaLM-Embedding-V2.5 | 494M | **71.29** | **65.31** | **90.50** | 58.12 | 86.63 | 47.42 | 58.45 | 84.82 | 31.21 |

Table 13: Detailed AIR-Bench QA results (NDCG@10 scores) on AIR benchmark 24.05. across seven languages.

| Model | Size | AIR-Bench QA | | | | | | | |
|---|---|---|---|---|---|---|---|---|---|
| | | MTK | en | zh | es | fr | ja | de | ru |
| bge-multilingual-gemma2 | 9B | 51.77 | 46.25 | 49.34 | 60.76 | 49.69 | 60.02 | 49.77 | 54.97 |
| gte-Qwen2-7B-instruct | 7B | 49.33 | 51.87 | 47.12 | 55.18 | 43.04 | 54.76 | 44.91 | 52.65 |
| gte-Qwen2-1.5B-instruct | 1.5B | 45.72 | 48.03 | 43.13 | 50.26 | 40.37 | 50.04 | 41.25 | 50.73 |
| jina-embeddings-v3 (Multi-LoRA) | 572M | 45.97 | 45.07 | 44.76 | 52.19 | 39.94 | 50.11 | 43.62 | 51.70 |
| multilingual-e5-large | 560M | 44.54 | 43.91 | 43.60 | 50.84 | 35.94 | 52.84 | 41.93 | 50.44 |
| bge-m3 (Dense) | 560M | **49.30** | 48.78 | 47.45 | 53.73 | **44.66** | 54.23 | **46.71** | **54.55** |
| KaLM-Embedding-V2.5 | 494M | 49.02 | **49.86** | **48.69** | **54.43** | 43.05 | 52.80 | 46.00 | 52.43 |

we observe that KaLM-Embedding-V2.5 demonstrates the superior discriminative capacity in all cases, while KaLM-Embedding-V1 and Qwen3-Embedding-0.6B perform poorly in the 1st and 3rd cases. Besides, the distance between the red star and the median (the green line) of the box plot for KaLM-Embedding-V2.5 is consistently larger than the corresponding distance for both KaLM-Embedding-V1 and Qwen3-Embedding-0.6B in most cases. This indicates that the distribution of their hard negative scores is too close to the positive, meaning their limited ability to distinguish subtle yet critical differences. The large and consistent margin maintained by KaLM-Embedding-V2.5 demonstrates the effectiveness of its improved training techniques, especially the Focal-style Reweighting Mechanism, which focuses on learning hard samples and leads to the large margin observed in the visualization. In conclusion, the qualitative results provide intuitive evidence that aligns with high quantitative benchmark performance, solidifying the model's effectiveness.

## F  VISUALIZATION ANALYSIS

To better understand the relationship between embedding quality and downstream task performance, we conduct a visualization analysis of different models on clustering and classification datasets, covering intent recognition, category identification, and topic classification, with both English and Chinese data included. As shown in Figure 6, we project embeddings into 2D by UMAP (Uniform Manifold Approximation and Projection), with colors indicating the corresponding labels of the data points. From the results, the embeddings produced by KaLM-Embedding-V2.5 exhibit more compact and separated clusters compared to KaLM-Embedding-V1 and Qwen3-Embedding-0.6B. In the RedditClustering and CLSClusteringP2P, semantically similar samples are tightly grouped under V2.5, while inter-class boundaries become more distinct, aligning with its superior clustering performance. In contrast, Qwen3-Embedding-0.6B displays overlapping regions between categories, suggesting a weaker capability in modeling fine-grained semantic distinctions. The results of the Banking77Classification further confirm this conclusion. KaLM-Embedding-V2.5 forms separated clusters, whereas V1 and Qwen3-Embedding-0.6B embeddings remain entangled. Overall, the improved intra-class compactness and inter-class separability of KaLM-Embedding-V2.5 provide strong support for its superior results on these tasks.

## G    MULTILINGUAL EVALUATION

To evaluate multilingual and OOD generalization, we adopt AIR-Bench QA (Chen et al., 2025b) (version of 24.05), which provides a more comprehensive test than static benchmarks. AIR-Bench is automatically generated to avoid data leakage, spans diverse tasks, domains, and languages. Table 13 shows the evaluation results on AIR-Bench QA, including seven languages: en (English), zh (Chinese), es (Spanish), fr (French), ja (Japanese), de (German), and ru (Russian). KaLM-Embedding-V2.5 is evaluated using a general retrieval instruction: *"Instruct: Given a query, retrieve documents that answer the query. \n Query: {query}"* Despite not being trained on large-scale multilingual corpora, KaLM-Embedding-V2.5 demonstrates competitive performance across all seven languages. It performs on par with or close to substantially larger 7B-9B models. For example, its average score (49.02) is nearly identical to that of the much larger gte-Qwen2-7B-instruct model (49.33). In lower-resource languages, its performance is comparable to strong multilingual embedding baselines, such as bge-m3. These results demonstrate that KaLM-Embedding-V2.5 generalizes well beyond its primary English-Chinese training focus, exhibiting robust retrieval performance across a wide range of multilingual and low-resource language settings.

## H    FULL MTEB RESULTS

Table 14 and Table 15 show the full METB results for each dataset.

## I    DATASETS AND INSTRUCTIONS

Table 16 and Table 17 show the detailed dataset list used for pre-training, and fine-tuning as well as distillation, respectively. Table 18 presents the task instructions used in the MTEB evaluation.

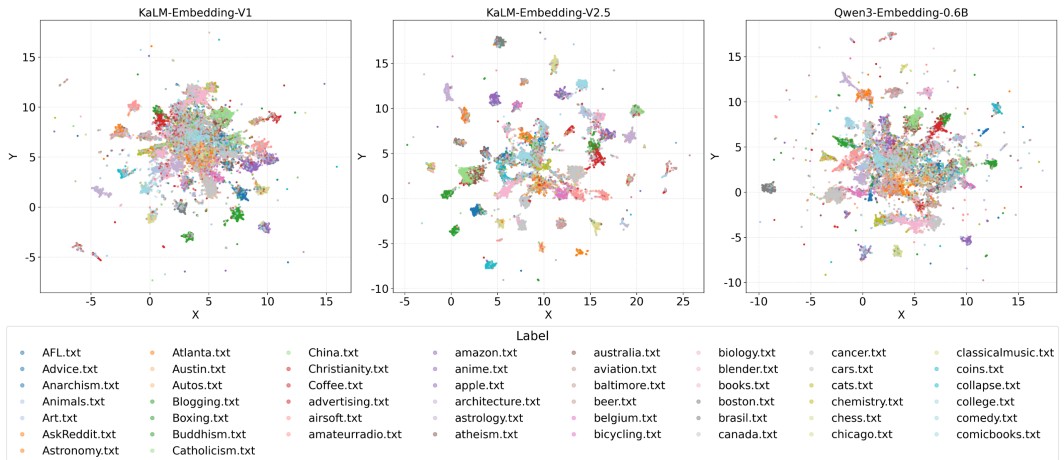

(a) RedditClustering, where the task instruction is "Instruct: Identify the topic or theme of Reddit posts based on the titles Query: {query }".

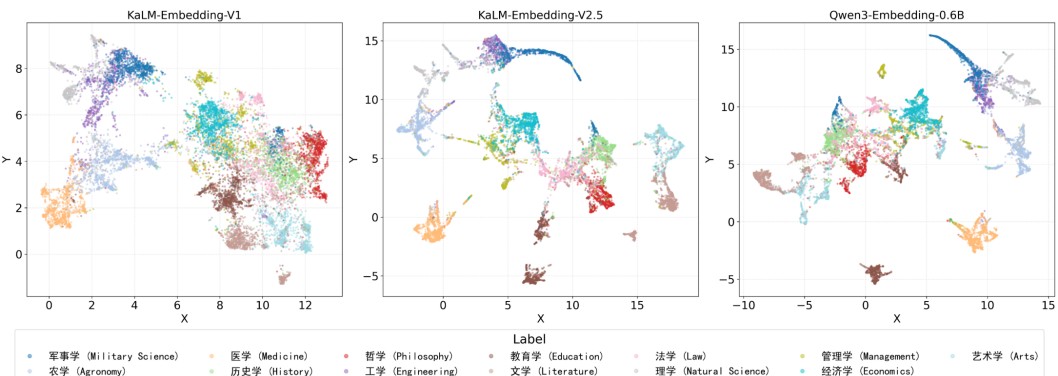

(b) CLSClusteringP2P, where the task instruction is "Instruct: Identify the main category of scholar papers based on the titles and abstracts Query: {query}".

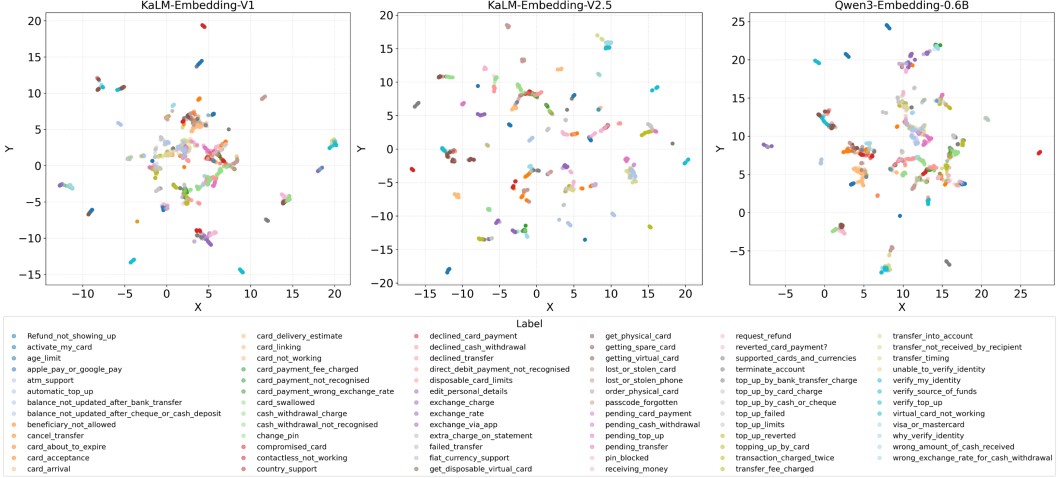

(c) Banking77Classification, where the task instruction is "Instruct: Given a online banking query, find the corresponding intents Query: {query}".

Figure 6: Embedding distribution comparisons between KaLM-Embedding-V1, KaLM-Embedding-V2.5, and Qwen3-Embedding-0.6B.

Table 14: Results for each dataset on MTEB (eng, v1). 'Emb' is the abbreviation of 'Embedding'

| | Dataset | KaLM-Emb-V1 | KaLM-Emb-V2 | KaLM-Emb-V2.5 |
|---|---|---|---|---|
| Classification | AmazonCounterfactualClassification | 91.73 | **95.25** | 94.75 |
| | AmazonPolarityClassification | 96.56 | 96.67 | **97.03** |
| | AmazonReviewsClassification | 61.42 | 57.89 | **64.15** |
| | Banking77Classification | 84.54 | 89.48 | **90.31** |
| | EmotionClassification | 86.90 | **92.50** | 83.80 |
| | ImdbClassification | 94.93 | 95.16 | **95.91** |
| | MassiveIntentClassification | 72.52 | 77.80 | **83.24** |
| | MassiveScenarioClassification | 79.32 | 86.00 | **89.35** |
| | MTOPDomainClassification | 97.54 | **98.86** | 98.69 |
| | MTOPIntentClassification | 85.76 | 88.77 | **91.10** |
| | ToxicConversationsClassification | 89.28 | 89.34 | **91.70** |
| | TweetSentimentExtractionClassification | 76.35 | 78.60 | **80.08** |
| Clustering | ArxivClusteringP2P | 49.68 | 51.16 | **52.11** |
| | ArxivClusteringS2S | 42.21 | 43.70 | **45.10** |
| | BiorxivClusteringP2P | 43.84 | 47.69 | **48.51** |
| | BiorxivClusteringS2S | 37.31 | 41.93 | **42.75** |
| | MedrxivClusteringP2P | 39.91 | **43.72** | 43.09 |
| | MedrxivClusteringS2S | 36.79 | **40.56** | 40.43 |
| | RedditClustering | 55.47 | 76.52 | **76.89** |
| | RedditClusteringP2P | 65.96 | **73.05** | 72.84 |
| | StackExchangeClustering | 66.38 | 78.40 | **80.22** |
| | StackExchangeClusteringP2P | 39.19 | 45.41 | **47.26** |
| | TwentyNewsgroupsClustering | 49.33 | **74.44** | 73.26 |
| Pair Classification | SprintDuplicateQuestions | 92.65 | 95.88 | **96.00** |
| | TwitterSemEval2015 | 71.44 | 76.72 | **77.15** |
| | TwitterURLCorpus | 85.69 | 85.95 | **86.66** |
| Reranking | AskUbuntuDupQuestions | 60.35 | 62.13 | **62.39** |
| | MindSmallReranking | 31.92 | 32.04 | **32.45** |
| | SciDocsRR | 80.99 | 82.25 | **84.68** |
| | StackOverflowDupQuestions | 48.38 | 50.54 | **51.82** |
| Retrieval | ArguAna | 58.63 | 57.42 | **60.15** |
| | ClimateFEVER | 25.85 | 25.07 | **34.50** |
| | CQADupstack | 41.83 | 44.19 | **47.20** |
| | DBPedia | 38.94 | 40.26 | **42.62** |
| | FEVER | 86.54 | 83.00 | **87.89** |
| | FiQA2018 | 44.74 | 45.23 | **47.10** |
| | HotpotQA | 67.58 | 70.14 | **71.76** |
| | MSMARCO | 34.59 | 36.20 | **40.62** |
| | NFCorpus | 35.33 | 35.17 | **37.11** |
| | NQ | 47.50 | 48.10 | **58.61** |
| | QuoraRetrieval | 87.47 | **89.81** | 89.57 |
| | SCIDOCS | 19.97 | 20.81 | **21.62** |
| | SciFact | 72.89 | 71.98 | **74.38** |
| | TRECCOVID | 83.72 | 79.27 | **82.98** |
| | Touche2020 | 29.15 | 28.43 | **28.93** |
| STS | BIOSSES | 86.14 | **84.16** | 84.02 |
| | SICK-R | 79.73 | 79.85 | **83.20** |
| | STS12 | 80.17 | **82.27** | 81.90 |
| | STS13 | 83.86 | 85.96 | **89.52** |
| | STS14 | 80.57 | 83.50 | **85.99** |
| | STS15 | 87.34 | 86.44 | **90.33** |
| | STS16 | 84.83 | 85.70 | **87.74** |
| | STS17 | 86.43 | 86.16 | **92.34** |
| | STS22 | 69.21 | 66.95 | **68.76** |
| | STSBenchmark | 84.12 | 85.07 | **88.88** |
| Summarization | SummEval | 25.23 | 28.51 | **31.18** |
| **Mean (Task)** | | 64.94 | 67.47 | **69.33** |
| **Mean (Type)** | | 61.49 | 64.14 | **65.83** |

Table 15: Results for each dataset on MTEB (cmn, v1).

| | Dataset | KaLM-Emb-V1 | KaLM-Emb-V2 | KaLM-Emb-V2.5 |
|---|---|---|---|---|
| Classification | IFlyTek | 48.54 | 51.01 | **56.59** |
| | JDReview | 83.02 | 86.87 | **88.82** |
| | MultilingualSentiment | 78.25 | 79.16 | **81.26** |
| | OnlineShopping | 93.08 | 94.40 | **95.02** |
| | TNews | 51.59 | 50.75 | **53.27** |
| | Waimai | 88.85 | 88.67 | **89.91** |
| Clustering | CLSClusteringP2P | 46.92 | 62.95 | **66.25** |
| | CLSClusteringS2S | 44.67 | 59.44 | **62.73** |
| | ThuNewsClusteringP2P | 72.87 | 80.79 | **84.64** |
| | ThuNewsClusteringS2S | 65.68 | 75.87 | **78.75** |
| Pair Classification | Cmnli | 76.67 | 78.08 | **86.07** |
| | Ocnli | 69.22 | 77.73 | **82.12** |
| Reranking | CMedQAv1-reranking | 82.34 | 83.65 | **84.58** |
| | CMedQAv2-reranking | 83.12 | 84.25 | **85.78** |
| | MMarcoReranking | 25.75 | 26.04 | **29.64** |
| | T2Reranking | 66.73 | 66.69 | **67.60** |
| Retrieval | CmedqaRetrieval | 42.12 | 44.81 | **45.87** |
| | CovidRetrieval | 82.40 | 83.30 | **83.57** |
| | DuRetrieval | 82.19 | 83.17 | **86.14** |
| | EcomRetrieval | 62.56 | 65.10 | **66.68** |
| | MedicalRetrieval | 56.89 | 59.81 | **60.46** |
| | MMarcoRetrieval | 78.96 | 80.59 | **82.23** |
| | T2Retrieval | 84.06 | 84.88 | **85.97** |
| | VideoRetrieval | 71.82 | 75.51 | **76.44** |
| STS | AFQMC | 38.02 | 44.18 | **48.78** |
| | ATEC | 46.19 | 49.75 | **52.45** |
| | BQ | 54.48 | 61.22 | **69.74** |
| | LCQMC | 70.81 | 73.83 | **77.50** |
| | PAWSX | 16.32 | 43.38 | **47.90** |
| | QBQTC | 35.28 | 37.61 | **39.83** |
| | STSB | 77.80 | 79.10 | **82.38** |
| **Mean (Task)** | | 63.78 | 68.15 | **70.93** |
| **Mean (Type)** | | 64.56 | 69.28 | **72.46** |

Table 16: Pre-training data list.

| Source | Language | Pairs |
|---|---|---|
| Amazon-Reviews (Hou et al., 2024) | multilingual | 23M |
| CC-News (Hamborg et al., 2017) | multilingual | 100M |
| NLLB (Costa-jussà et al., 2022; Heffernan et al., 2022; Schwenk et al., 2021) | multilingual | 2M |
| Wikipedia (Foundation, 2024) | multilingual | 100M |
| xP3 (Muennighoff et al., 2023b) | multilingual | 19M |
| XL-Sum (Hasan et al., 2021) | multilingual | 1M |
| SWIM-IR (Monolingual) (Thakur et al., 2024) | multilingual | 3M |
| SWIM-IR (Cross-lingual) (Thakur et al., 2024) | multilingual | 15M |
| CSL (Li et al., 2022) | zh | 0.4M |
| Wudao (Yuan et al., 2021) | zh | 44M |
| THUCNews (Sun et al., 2016) | zh | 0.8M |
| Zhihu-KOL | zh | 0.8M |
| CodeSearchNet (Husain et al., 2019) | en | 1M |
| PAQ (Lewis et al., 2021) | en | 9M |
| Reddit | en | 100M |
| StackExchange | en | 14M |
| S2ORC | en | 41M |

Table 17: Fine-tuning data list.

| Source | Type | Categ. | Language | Pairs | Pairs(filtered) |
|---|---|---|---|---|---|
| CodeFeedback (Zheng et al., 2024) | Retrieval | s2p | en | 50000 | 49090 |
| ELI5 (Fan et al., 2019) | Retrieval | s2p | en | 100000 | 76408 |
| ExpertQA (Malaviya et al., 2024) | Retrieval | s2p | en | 1261 | 1252 |
| GooAQ (Khashabi et al., 2021) | Retrieval | s2p | en | 50000 | 49833 |
| MEDI2BGE (Muennighoff et al., 2024; Su et al., 2023) | Retrieval | s2p | en | 100000 | 71790 |
| OpenOrca (Mukherjee et al., 2023) | Retrieval | s2p | en | 40000 | 38623 |
| PAQ (Lewis et al., 2021) | Retrieval | s2p | en | 50000 | 49849 |
| PubMedQA (Jin et al., 2019) | Retrieval | s2p | en | 80000 | 79954 |
| SearchQA (Dunn et al., 2017) | Retrieval | s2p | en | 10000 | 9988 |
| arxiv_qa | Retrieval | s2p | en | 23397 | 17927 |
| CC-News (Hamborg et al., 2017) | Retrieval | s2p | en | 30000 | 28246 |
| TREC-COVID (Voorhees et al., 2020; Wang et al., 2020) | Retrieval | s2p | en | 50000 | 48517 |
| DBpedia-Entity (Thakur et al., 2021) | Retrieval | s2p | en | 100000 | 96792 |
| ESCI (Reddy et al., 2022) | Retrieval | s2p | en | 30000 | 26043 |
| FEVER (Thorne et al., 2018) | Retrieval | s2p | en | 87855 | 87216 |
| FiQA (Maia et al., 2018) | Retrieval | s2p | en | 5490 | 4689 |
| HotpotQA (Yang et al., 2018) | Retrieval | s2p | en | 184057 | 150153 |
| MLDR (Chen et al., 2024) | Retrieval | s2p | en | 41434 | 31097 |
| MSMARCO (Nguyen et al., 2016) | Retrieval | s2p | en | 175133 | 174190 |
| MSMARCO-v2 (Nguyen et al., 2016) | Retrieval | s2p | en | 277144 | 258617 |
| NFCorpus (Boteva et al., 2016) | Retrieval | s2p | en | 10824 | 10471 |
| rag-dataset-12000 | Retrieval | s2p | en | 9590 | 9272 |
| SciFact (Wadden et al., 2020) | Retrieval | s2p | en | 809 | 794 |
| SQuAD 2.0 (Rajpurkar et al., 2018; 2016) | Retrieval | s2p | en | 130217 | 125816 |
| TriviaQA (Joshi et al., 2017) | Retrieval | s2p | en | 52886 | 44442 |
| WebGPT Comparisons (Nakano et al., 2021) | Retrieval | s2p | en | 19242 | 18924 |
| Natural Questions (Kwiatkowski et al., 2019) | Retrieval | s2p | en | 58622 | 56377 |
| Yahoo Answers | Retrieval | s2p | en | 30000 | 21724 |
| CQADupStack (Hoogeveen et al., 2015) | Retrieval | s2p | en | 24045 | 7356 |
| ContractNLI (Koreeda & Manning, 2021) | STS | s2s | en | 3195 | 628 |
| MultiNLI (Williams et al., 2018) | STS | s2s | en | 64674 | 63701 |
| NLLB (Costa-jussà et al., 2022; Heffernan et al., 2022) | STS | s2s | en | 36000 | 26504 |
| Quora (DataCanary, 2017) | STS | s2s | en | 92674 | 89558 |
| WikiAnswers (Fader et al., 2014) | STS | s2s | en | 50000 | 47686 |
| SimCSE NLI (Gao et al., 2021) | STS | s2s | en | 252397 | 217099 |
| SNLI (Bowman et al., 2015) | STS | s2s | en | 24686 | 16480 |
| arXiv | Classfication | s2s, p2s | en | 15000 | 14529 |
| Biorxiv | Classfication | s2s, p2s | en | 6862 | 6787 |
| Medrxiv | Classfication | s2s, p2s | en | 2012 | 1999 |
| Reddit-Clustering (Geigle et al., 2021) | Classfication | s2s | en | 128000 | 25600 |
| Reddit-Clustering-P2P (Reimers, 2021) | Classfication | p2s | en | 12704958 | 42480 |
| Stackexchange-Clustering (Geigle et al., 2021) | Classfication | s2s | en | 1014826 | 50530 |
| Stackexchange-Clustering-P2P (Stack Exchange, 2021) | Classfication | p2s | en | 25333327 | 48800 |
| TwentyNewsgroups-Clustering (Lang, 1995) | Classfication | s2s | en | 11314 | 6233 |
| AmazonPolarity (McAuley & Leskovec, 2013) | Classfication | s2s | en | 10000 | 9007 |
| IMDB (Maas et al., 2011) | Classfication | s2s | en | 10000 | 8575 |
| banking77 (Casanueva et al., 2020) | Classfication | s2s | en | 10000 | 9937 |
| EmotionClassification (Saravia et al., 2018) | Classfication | s2s | en | 10000 | 10000 |
| TweetSentimentExtraction | Classfication | s2s | en | 10000 | 10000 |
| ToxicConversations | Classfication | s2s | en | 7916 | 7800 |
| AdvertiseGen (Shao et al., 2019) | Retrieval | s2p | zh | 20000 | 17526 |
| CHEF (Hu et al., 2022) | Retrieval | s2p | zh | 4952 | 4824 |
| ChatMed-Dataset (Zhu, 2023) | Retrieval | s2p | zh | 20000 | 18608 |
| CMRC 2018 (Cui et al., 2019) | Retrieval | s2p | zh | 10000 | 9753 |
| DRCD (Shao et al., 2018) | Retrieval | s2p | zh | 5000 | 4714 |
| LCSTS (Hu et al., 2015) | Retrieval | s2p | zh | 20000 | 19535 |
| LIMA (Zhou et al., 2023) | Retrieval | s2p | zh | 2058 | 1991 |
| Multi-CPR (Long et al., 2022) | Retrieval | s2p | zh | 287881 | 234587 |
| PAWS-X (zh) (Yang et al., 2019) | Retrieval | s2p | zh | 49401 | 19289 |
| RefGPT (Yang et al., 2023) | Retrieval | s2p | zh | 50000 | 49896 |
| T2Ranking (Xie et al., 2023) | Retrieval | s2p | zh | 199412 | 188606 |
| THUCNews (Sun et al., 2016) | Retrieval | s2p | zh | 20000 | 19288 |
| UMETRIP-QA | Retrieval | s2p | zh | 2647 | 2537 |
| WebCPM (Qin et al., 2023) | Retrieval | s2p | zh | 1605 | 1602 |
| cCOVID-News | Retrieval | s2p | zh | 5000 | 4727 |
| cMedQA-V2.0 (Zhang et al., 2018) | Retrieval | s2p | zh | 223851 | 88109 |
| CSL (Li et al., 2022) | Retrieval | s2p | zh | 20000 | 19945 |
| DuReader (He et al., 2018) | Retrieval | s2p | zh | 80416 | 79229 |
| DuReader_checklist (Tang et al., 2021) | Retrieval | s2p | zh | 99992 | 97764 |
| law-gpt (Liu et al., 2023) | Retrieval | s2p | zh | 500 | 500 |
| lawzhidao (Ustinian, 2020) | Retrieval | s2p | zh | 8000 | 6784 |
| mMARCO (zh) (Bonifacio et al., 2021) | Retrieval | s2p | zh | 400000 | 379870 |
| retrieval_data_llm | Retrieval | s2p | zh | 32768 | 32551 |
| webqa | Retrieval | s2p | zh | 5000 | 4988 |
| AFQMC | STS | s2s | zh | 4041 | 3876 |
| ATEC | STS | s2s | zh | 62477 | 11387 |
| BQ | STS | s2s | zh | 100000 | 10000 |
| CAIL2019-SCM (Xiao et al., 2019) | STS | s2s | zh | 5102 | 648 |
| CINLID | STS | s2s | zh | 5000 | 2883 |
| ChineseSTS (Tang et al., 2016) | STS | s2s | zh | 2500 | 2497 |
| CMNLI (Xu et al., 2020) | STS | s2s | zh | 125356 | 119029 |
| nli_zh (Chen et al., 2018; Liu et al., 2018a; Yang et al., 2019) | STS | s2s | zh | 218887 | 185787 |
| OCNLI (Hu et al., 2020) | STS | s2s | zh | 13464 | 11937 |
| QBQTC | STS | s2s | zh | 51620 | 47223 |
| SimCLUE | STS | s2s | zh | 344038 | 290699 |
| XNLI (zh) (Conneau et al., 2018) | STS | s2s | zh | 80000 | 74252 |
| CSL (Li et al., 2022) | Classfication | s2s, p2s | zh | 15000 | 12249 |
| THUCNews (Sun et al., 2016) | Classfication | s2s | zh | 10000 | 9690 |
| TNews | Classfication | s2s | zh | 10000 | 6762 |
| JDReview | Classfication | s2s | zh | 1232 | 1232 |
| IFlyTek (Zhao et al., 2022) | Classfication | s2s | zh | 10000 | 8221 |
| OnlineShopping | Classfication | s2s | zh | 7852 | 7600 |
| Waimai | Classfication | s2s | zh | 7384 | 7376 |
| Aya Dataset (Singh et al., 2024) | Retrieval | s2p | multilingual | 30000 | 26292 |
| MIRACL (Zhang et al., 2023) | Retrieval | s2p | multilingual | 40151 | 39946 |
| Mr. TyDi (Zhang et al., 2021) | Retrieval | s2p | multilingual | 48729 | 46997 |
| PAWS-X (Yang et al., 2019) | STS | s2s | multilingual | 128435 | 128398 |
| AmazonReviews (Ni et al., 2019) | Classfication | s2s | multilingual | 10000 | 7721 |
| AmazonCounterfactual (O'Neill et al., 2021) | Classfication | s2s | multilingual | 10000 | 8323 |
| MultilingualSentiment (Mollanorozy et al., 2023) | Classfication | s2s | multilingual | 10000 | 9804 |
| Amazon Massive Intent (FitzGerald et al., 2023) | Classfication | s2s | multilingual | 10000 | 7832 |
| AmazonMassiveScenario (FitzGerald et al., 2023) | Classfication | s2s | multilingual | 10000 | 7078 |
| MTOPDomain (Li et al., 2021) | Classfication | s2s | multilingual | 10000 | 9610 |
| MTOPIntent (Li et al., 2021) | Classfication | s2s | multilingual | 10000 | 7952 |

Table 18: Detailed task instruction list for MTEB evaluation. Pair Classification*, Reranking*, Retrieval*, and STS* indicate we use the same instructions for all the respective remaining tasks.

| Task Name | Instruction |
|---|---|
| **Classification** | |
| AmazonCounterfactualClassification | Instruct: Given an Amazon review, judge whether it is counterfactual. \n Query: {query} |
| AmazonPolarityClassification | Instruct: Classifying Amazon reviews into positive or negative sentiment \n Query: {query} |
| AmazonReviewsClassification | Instruct: Classifying the given Amazon review into its appropriate rating category \n Query: {query} |
| Banking77Classification | Instruct: Given a online banking query, find the corresponding intents \n Query: {query} |
| EmotionClassification | Instruct: Classifying the emotion expressed in the given Twitter message into one of the six emotions: anger, fear, joy, love, sadness, and surprise \n Query: {query} |
| ImdbClassification | Instruct: Classifying the sentiment expressed in the given movie review text from the IMDB dataset \n Query: {query} |
| MassiveIntentClassification | Instruct: Given a user utterance as query, find the user intents \n Query: {query} |
| MassiveScenarioClassification | Instruct: Given a user utterance as query, find the user scenarios \n Query: {query} |
| MTOPDomainClassification | Instruct: Classifying the intent domain of the given utterance in task-oriented conversation \n Query: {query} |
| MTOPIntentClassification | Instruct: Classifying the intent of the given utterance in task-oriented conversation \n Query: {query} |
| ToxicConversationsClassification | Instruct: Classifying the given comments as either toxic or not toxic \n Query: {query} |
| TweetSentimentExtractionClassification | Instruct: Classifying the sentiment of a given tweet as either positive, negative, or neutral \n Query: {query} |
| TNews | Instruct: Categorizing the given news title \n Query: {query} |
| IFlyTek | Instruct: Given an App description text, find the appropriate fine-grained category \n Query: {query} |
| MultilingualSentiment | Instruct: Classifying sentiment of the customer review into positive, neutral, or negative \n Query: {query} |
| JDReview | Instruct: Classifying sentiment of the customer review for iPhone into positive or negative \n Query: {query} |
| OnlineShopping | Instruct: Classifying sentiment of the customer review into positive or negative \n Query: {query} |
| Waimai | Instruct: Classify the customer review from a food takeaway platform into positive or negative \n Query: {query} |
| **Clustering** | |
| ArxivClusteringP2P | Instruct: Identify the main and secondary category of Arxiv papers based on the titles and abstracts \n Query: {query} |
| ArxivClusteringS2S | Instruct: Identify the main and secondary category of Arxiv papers based on the titles \n Query: {query} |
| BiorxivClusteringP2P | Instruct: Identify the main category of Biorxiv papers based on the titles and abstracts \n Query: {query} |
| BiorxivClusteringS2S | Instruct: Identify the main category of Biorxiv papers based on the titles \n Query: {query} |
| MedrxivClusteringP2P | Instruct: Identify the main category of Medrxiv papers based on the titles and abstracts \n Query: {query} |
| MedrxivClusteringS2S | Instruct: Identify the main category of Medrxiv papers based on the titles \n Query: {query} |
| RedditClustering | Instruct: Identify the topic or theme of Reddit posts based on the titles \n Query: {query} |
| RedditClusteringP2P | Instruct: Identify the topic or theme of Reddit posts based on the titles and posts \n Query: {query} |
| StackExchangeClustering | Instruct: Identify the topic or theme of StackExchange posts based on the titles \n Query: {query} |
| StackExchangeClusteringP2P | Instruct: Identify the topic or theme of StackExchange posts based on the given paragraphs \n Query: {query} |
| TwentyNewsgroupsClustering | Instruct: Identify the topic or theme of the given news articles \n Query: {query} |
| CLSClusteringS2S | Instruct: Identify the main category of scholar papers based on the titles \n Query: {query} |
| CLSClusteringP2P | Instruct: Identify the main category of scholar papers based on the titles and abstracts \n Query: {query} |
| ThuNewsClusteringS2S | Instruct: Identify the topic or theme of the given news articles based on the titles \n Query: {query} |
| ThuNewsClusteringP2P | Instruct: Identify the topic or theme of the given news articles based on the titles and contents \n Query: {query} |
| **Pair Classification** | |
| Pair Classification* | Instruct: Retrieve semantically similar text. \n Query: {query} |
| SprintDuplicateQuestions | Instruct: Retrieve semantically similar questions. \n Query: {query} |
| **Reranking** | |
| Reranking* | Instruct: Given a query, retrieve documents that answer the query. \n Query: {query} |
| AskUbuntuDupQuestions | Instruct: Retrieve semantically similar questions. \n Query: {query} |
| StackOverflowDupQuestions | Instruct: Retrieve semantically similar questions. \n Query: {query} |
| SciDocsRR | Instruct: Retrieve relevant paper titles \n Query: {query} |
| **Retrieval** | |
| Retrieval* | Instruct: Given a query, retrieve documents that answer the query. \n Query: {query} |
| QuoraRetrieval | Instruct: Retrieve semantically similar questions. \n Query: {query} |
| CQADupstack | Instruct: Given a question, retrieve detailed question descriptions from Stackexchange that are duplicates to the given question \n Query: {query} |
| **STS** | |
| STS* | Instruct: Retrieve semantically similar text. \n Query: {query} |
| **Summarization** | |
| SummEval | Instruct: Retrieve semantically similar summaries. \n Query: {query} |

