# OpenReview forum: "KaLM-Embedding-V2: Superior Training Techniques and Data Inspire A Versatile Embedding Model"
_ICLR.cc/2026/Conference — ICLR 2026 Poster_

### Official Review · Reviewer_kXYw · 2025-10-26

**Soundness:** 3
**Presentation:** 3
**Contribution:** 3
**Rating:** 6
**Confidence:** 4

**Summary:**

This paper presents KaLM-Embedding-V2, a compact (0.5B parameters) text embedding model trained with improved data curation and training techniques.

**Strengths:**

1. The model achieves state-of-the-art results among models under 1B parameters and performs competitively with much larger models.
2. The paper includes detailed ablation studies, comparisons, and analyses across multiple benchmarks, which make the results convincing.
3. Open-source release

**Weaknesses:**

The contribution is mainly the integration of multiple known techniques, rather than a fundamentally novel algorithm.

**Questions:**

1. The teacher model is Qwen3-Embedding-8B. Why did the authors choose to initialize KaLM-Embedding-V2 from Qwen2-0.5B instead of using an existing Qwen-Embedding model (e.g., Qwen-Embedding-0.6B)? Wouldn’t that potentially eliminate the need for the expensive pre-training stage? Have the authors compared starting from Qwen2-base versus Qwen-Embedding models to verify whether the pre-training stage is truly necessary or beneficial?
2. Whether the proposed distillation strategy generalizes across different teacher families or if it only works within the Qwen series.

---

> ### Author Response · Authors · 2025-11-21
> **(1/2) Response to Reviewer kXYw**
>
> We sincerely thank the reviewer for the constructive comments and suggestions, which are very helpful for improving our paper. We are also grateful that you recognised the strengths of our paper. We have updated the manuscript accordingly and kindly invite the reviewer to check the revised version for details. Please kindly find point-to-point responses below.
> > Weakness 1: The contribution is mainly the integration of multiple known techniques, rather than a fundamentally novel algorithm.
>
> We sincerely thank the reviewer for raising this important point. While our work leverages some common techniques in embedding training (e.g., multi-stage training), it introduces systematic innovations across multiple dimensions:
> - Model architecture: a series of versatile and compact embedding models with fully bidirectional attention tailored for representation learning;
> - Training recipe: a progressive multi-stage pipeline from weakly-supervised pre-training→supervised fine-tuning→fine-grained contrastive distillation;
> - Training techniques: focal-style reweighting to emphasize difficult samples, synergized with online hard-negative mixing to amplify learning signals;
> - Data engineering: curated datasets with 20+ pre-training and 100+ fine-tuning/distillation categories, fully integrated into a reproducible data recipe.
>
> Moreover, we fully open-source models, code, and data, enabling reproducibility and reducing the barrier for building industrial-level text embeddings. We hope this clarifies the novelty and value of our work.

---

> ### Author Response · Authors · 2025-11-21
> **(2/2) Response to Reviewer kXYw**
>
> > Question 1: The teacher model is Qwen3-Embedding-8B. Why did the authors choose to initialize KaLM-Embedding-V2 from Qwen2-0.5B instead of using an existing Qwen-Embedding model (e.g., Qwen-Embedding-0.6B)? Wouldn’t that potentially eliminate the need for the expensive pre-training stage? Have the authors compared starting from Qwen2-base versus Qwen-Embedding models to verify whether the pre-training stage is truly necessary or beneficial?
>
> We sincerely thank the reviewer for this insightful question. Our choice of both the teacher model and the initialization strategy is driven by the goal of developing a fully open, reproducible, end-to-end recipe for training high-quality embedding models.
>
> First, we adopt Qwen3-Embedding-8B as the teacher model because it is widely recognized as one of the strongest publicly available embedding models. Using a high-quality teacher ensures that the distilled soft signals are reliable and sufficiently informative for fine-grained contrastive distillation.
>
> Regarding initialization, while models like Qwen-Embedding-0.6B are indeed strong candidates, we intentionally choose Qwen2-0.5B (base language model) as the starting point for three reasons:
>
> - We aim to provide a comprehensive and transparent training pipeline—from base LM→ weakly-supervised pre-training→supervised fine-tuning→contrastive distillation.
> - Using a base LM allows us to systematically analyze and optimize every stage of the pipeline, including architecture design, data engineering, training recipes, and training techniques.
> - The technique is designed to be generalizable, compatible with any base language model, and thus also applicable to existing embedding models.
>
> To address whether the pre-training stage is necessary, we conducted an ablation study in which we removed the weakly-supervised pre-training stage and directly fine-tuned the model. The results below show that the performance drop is considerable.
> | Setting             | MTEB (eng, v1) | MTEB (cmn, v1) |
> |--------------------|----------------|----------------|
> | KaLM-Embedding-V2.5 | 69.33          | 70.93          |
> | w/o pretraining      | 68.61 (−0.72)  | 70.65 (−0.28)  |
>
>
> These results demonstrate that the pre-training stage contributes essential general-purpose representation capabilities. We hope this clarifies the motivation behind our initialization strategy and the importance of the pre-training stage.
>
> > Quesion 2: Whether the proposed distillation strategy generalizes across different teacher families or if it only works within the Qwen series.
>
> We sincerely thank the reviewer for this insightful question. The proposed contrastive distillation method is model-agnostic and is not tied to the Qwen family.
>
> Our approach distills the normalized distribution of temperature-scaled cosine-similarity scores, instead of distilling hidden features. Because we only use these task-level similarity scores, the approach does not depend on the teacher's architecture or embedding space. This means the distillation strategy can work with any teacher model, not just the Qwen family.
>
> Although our experiments use Qwen3-Embedding-8B as the teacher, the method itself is fundamentally general and can be readily applied to other teacher model families. We have clarified this point in the revised manuscript.

---

> ### Comment · Reviewer_kXYw · 2025-11-26
>
> Thank you for the rebuttal and the updated manuscript. While the clarifications are appreciated, several key concerns remain.
>
> The question of initialization is still unresolved, as there is no comparison between starting from a base LM and from an existing embedding model, making the motivation unconvincing. The benefit of the pre-training stage also appears marginal, and no additional evidence is provided to justify its cost. Overall, the contribution still reads as a collection of incremental techniques rather than a clearly supported systematic innovation. The empirical results are practically useful, so I maintain my original score.

---

> > ### Author Response · Authors · 2025-11-27
> > **Response to Reviewer kXYw**
> >
> > We sincerely thank the reviewer for the constructive and practical feedback. To address your concerns, we are conducting additional experiments comparing models initialized from different base language models versus existing embedding models. We believe these results will clarify the impact of initialization choices and further support the motivation of our approach.

---

> ### Author Response · Authors · 2025-12-03
> **(2/2) Response to Reviewer kXYw  (Supplement)**
>
> We sincerely thank the reviewer for the constructive feedback. Regarding the reviewer's concern about the initialization issue, we have conducted additional experiments to address this concern. In the original version, KaLM-Embedding-V2 was initialized from Qwen2-0.5B, followed by pretraining and finetuning.
>
> Following the reviewer's suggestion, we further included experiments using Qwen3-0.6B as the initialization model, followed by pretraining and finetuning. The new model, KaLM-Embedding-V2 (Qwen3-0.6B), was trained with settings similar to Qwen3-Embedding-0.6B, including causal attention and term masking. The experimental results on MTEB (cmn, v1) are shown below:
>
> | Models                               | Size | MTY   | MTY   | Classification | Clustering | PairClassification | Reranking | Retrieval | STS   |
> |--------------------------------------|------|-------|-------|----------------|------------|---------------------|-----------|-----------|-------|
> | Qwen3-Embedding-0.6B            | 560M | 66.33 | 67.44 | 71.40          | 68.74      | 76.42               | 62.58     | 71.03     | 54.52 |
> | KaLM-Embedding-V2 (Qwen3-0.6B)       | 560M | 66.97 | 67.36 | **76.11**         | 65.91     | 70.89              | **65.45**    | 71.78    | 54.00 |
> | KaLM-Embedding-V2 (Qwen2-0.5B)       | **494M** | **68.15** | **69.28** | 75.14         | **69.76**     | **77.91**              | 65.16    | **72.15**    | **55.58** |
> |                |   |    |  |    |   |   |   |  |   |
> |                |   |    |  |    |   |   |   |  |   |
> |        KaLM-Embedding-V2.5 (Qwen2-0.5B)        | 494M  |  70.93  | 72.46 |  77.48  | 73.09  | 84.09  | 66.90  | 73.42 | 59.80  |
>
> The experimental results show that KaLM-Embedding-V2 (Qwen2-0.5B) consistently outperforms both KaLM-Embedding-V2 (Qwen3-0.6B) and Qwen3-Embedding-0.6B across most tasks, despite having fewer parameters. This indicates that while the choice of initialization model does influence performance, it is only one of the contributing factors in building a high-quality embedding model. The superior performance of the KaLM-Embedding-V2 series stems from the synergistic effect of superior architecture design, training techniques, training recipes, and data engineering.

---

### Official Review · Reviewer_4xdV · 2025-10-29

**Soundness:** 3
**Presentation:** 2
**Contribution:** 2
**Rating:** 4
**Confidence:** 3

**Summary:**

This paper presents KaLM-Embedding-V2, an embedding model that leverages Qwen2-0.5B and different training tricks to improve the embedding representation and accuracy on MTEB tasks. The model surpasses most of the selected small models and even selected larger models on MTEB English and Chinese v1.

**Strengths:**

1 - New embedding models that surpass some SoTA LLM-based and encoder-based models with very few parameters.

2 - Authors give extensive details about the training recipe and the datasets used for it.

3 - The model is compared to multiple existing models and evaluated on both English and Chinese.

4 - Good open-source contribution if the training code is released, as it goes beyond most recent models that are open-weights only.

**Weaknesses:**

1 - The originality of this work is not very clear, as it combines existing training recipes that were used for training embedding models (see NV-Embed models and Qwen3 embedding models).

2 - Table 2 is clearly missing SoTA models on MTEB(eng, v1). Checking the MTEB leaderboard (on MTEB(eng, v2)), I see for example that Qwen3-4B and 8B achieve good performance on the benchmark but are not listed in the >1B models list.  They outperform KaLM-v2 but they are larger, it would be great to list them.

3 - Table 5 is very dense and hard to read, probably a plot of the average score on MTEB with the ablations could be better.

4 - The model is fine-tuned on many training sets of the datasets that are used for MTEB evaluation. MTEB uses the test sets, but there is a distribution similarity between the sets that may help the model get better on the benchmark. Evaluating on MTEB(eng, v2) could have fixed some of these issues, as for example it removes MS MARCO that is used for this model finetuning. Also, evaluating on AIRBench could give more insights on how the model performs on out-of-domain and out-of-distribution data (see NV-embed paper)

5 - The model is not evaluated in a multilingual setting (2 high resource languages are not enough) but is compared to multilingual models. It would be great to add a multilingual evaluation.

**Questions:**

1 - How does the model perform on other languages than English and Chinese?

2 - Why not evaluate on MTEB(eng, v2) [1]?


[1] Enevoldsen, Kenneth, et al. "Mmteb: Massive multilingual text embedding benchmark." arXiv preprint arXiv:2502.13595 (2025).

---

> ### Author Response · Authors · 2025-11-21
> **(1/3) Response to Reviewer 4xdV**
>
> We sincerely thank the reviewer for the constructive comments and suggestions, which are very helpful for improving our paper. We are also grateful that you recognised the strengths of our paper. We have updated the manuscript accordingly and kindly invite the reviewer to check the revised version for details. Please kindly find point-to-point responses below.
>
> > Weakness 1:  The originality of this work is not very clear, as it combines existing training recipes that were used for training embedding models (see NV-Embed models and Qwen3 embedding models).
>
> We sincerely thank the reviewer for raising this important point and would like to clarify the originality and contributions of our work. We highly recognize the strong contributions of NV-Embed and Qwen3-Embedding, both of which have significantly advanced the field of text embeddings.
>
> While multi-stage training is a common practice in embedding models, our work was developed contemporaneously with Qwen3-Embedding, and the design differs fundamentally. For example, Qwen3-Embedding adopts pre-training→fine-tuning→model soup. In contrast, our pipeline uses weakly-supervised pre-training→supervised fine-tuning→fine-grained contrastive distillation, with progressively refined signals absent from both Qwen3-Embedding and NV-Embed.
>
> Beyond common practices, our work provides a systematic innovation across model architecture, training recipe, training techniques, and data engineering: (1) model architecture, which enables fully bidirectional attention; (2) training recipe, which introduces a progressive multi-stage training pipeline; (3) training techniques, which design a focal-style reweighting mechanism to emphasize difficult samples and enrich hard negatives via online hard-negative mixing; and (4) data engineering, which curate over 20 categories of data for pre-training and 100 categories of data for fine-tuning, fully integrated into a reproducible recipe.
>
> Unlike NV-Embed and Qwen3-Embedding, which only release the model weights, we fully open-source models, code, and data, enabling complete reproducibility and significantly lowering the barrier for academic research to build competitive text embeddings. We sincerely thank the reviewer 4xdV once again for the thoughtful feedback. We hope these clarifications address the concerns regarding originality and clearly demonstrate the novelty and value of our work.
>
> >Weakness 2: Table 2 is clearly missing SoTA models on MTEB(eng, v1). Checking the MTEB leaderboard (on MTEB(eng, v2)), I see for example that Qwen3-4B and 8B achieve good performance on the benchmark but are not listed in the >1B models list. They outperform KaLM-v2 but they are larger, it would be great to list them.
>
> Thank you for your constructive comment. We have included the results of Qwen3-Embedding-8B and 4B in the revised manuscript. As expected, KaLM-Embedding-V2.5 performs below these larger models, since it uses only 1/16 (vs. 8B) and 1/8 (vs. 4B) of their parameters.
>
> > Weakness 3: Table 5 is very dense and hard to read, probably a plot of the average score on MTEB with the ablations could be better.
>
> Thank you for the helpful suggestion. We agree that Table 5 is dense and that a visual presentation can improve readability. Following your suggestion, we have added a plot illustrating the average MTEB score across ablation settings in the revised manuscript in Figure 4. This visualization makes the performance trends clearer and complements the detailed numerical results in Table 5.

---

> ### Author Response · Authors · 2025-11-21
> **(2/3) Response to Reviewer 4xdV**
>
> > Weakness 4: The model is fine-tuned on many training sets of the datasets that are used for MTEB evaluation. MTEB uses the test sets, but there is a distribution similarity between the sets that may help the model get better on the benchmark. Evaluating on MTEB(eng, v2) could have fixed some of these issues, as for example it removes MS MARCO that is used for this model finetuning. Also, evaluating on AIRBench could give more insights on how the model performs on out-of-domain and out-of-distribution data (see NV-embed paper)
>
> > Question 2: Why not evaluate on MTEB(eng, v2) [1]?
>
> Thank you for raising this important point. We fully agree that MTEB (eng, v2) resolves several known issues in MTEB (eng, v1). Following your suggestion, we re-evaluated our model on MTEB (eng, v2) and report the full results below. We compare against strong baselines including Qwen3-Embedding-8B/4B/0.6B, multilingual-e5-large-instruct, bge-large-en-v1.5, and embedding-gemma-300M. KaLM-Embedding-V2.5 achieves the best overall performance among models of comparable scale. We appreciate the reviewer 4xdV's recommendation, which significantly enhances the quality of our paper.
>
> | Models                          | Size  | MTY   | MTY   | Classification | Clustering | PairClassification | Reranking | Retrieval | STS   | Summarization |
> |---------------------------------|-------|-------|-------|----------------|------------|------------------|-----------|-----------|-------|---------------|
> | Qwen3-Embedding-8B               | 8B    | 75.22 | 68.70 | 90.43          | 58.57      | 87.52            | 51.56     | 69.44     | 88.58 | 34.83         |
> | Qwen3-Embedding-4B               | 4B    | 74.60 | 68.09 | 89.84          | 57.51      | 87.01            | 50.76     | 68.46     | 88.72 | 34.39
> | |  | |   |           |        |           |     |      |   |         |
> | Qwen3-Embedding-0.6B             | 596M  | $\underline{\text{70.70}}$ | 64.88 | 85.76          | 54.05      | 84.37            | 48.18     | **61.83**     | **86.57** | $\underline{\text{33.43}}$         |
> | multilingual-e5-large-instruct   | 560M  | 65.53 | 61.21 | 75.54          | 49.89      | 86.24            | **48.74**     | 53.47     | 84.72 | 29.89         |
> | bge-large-en-v1.5                | 335M  | 65.89 | 61.87 | 78.34          | 48.01      | $\underline{\text{87.13}}$            | $\underline{\text{48.26}}$     | 55.44     | 82.79 | 33.13         |
> | embeddinggemma-300m               | 307M  | 69.67 | $\underline{\text{65.11}}$ | $\underline{\text{87.55}}$          | $\underline{\text{56.55}}$      | **87.29**            | 47.43     | 55.69     | 83.61 | **37.64**         |
> | KaLM-Embedding-V2.5               | 494M  | **71.29** | **65.31** | **90.50**          | **58.12**      | 86.63            | 47.42     | $\underline{\text{58.45}}$   | $\underline{\text{84.82}}$ | 31.21         |
>
> To further examine out-of-domain (OOD) generalization, we evaluated our model on AIRBench QA, including English (en) and Chinese (zh). We used the general retrieval instruction for KaLM-Embedding-V2.5: *Given a query, retrieve documents that answer the query \n Query: {query}*. Notably, Qwen3-Embedding-0.6B is not included because no official AIRBench results are available. On AIRBench, KaLM-Embedding-V2.5 shows strong OOD performance, outperforming bge-multilingual-gemma2 and matching the performance of gte-Qwen2-7B-instruct.
>
> | Models                    | Size  | MYK  | en    | zh    |
> |---------------------------|-------|------|-------|-------|
> | bge-multilingual-gemma2   | 9B    | 47.53| 46.25 | 49.34 |
> | gte-Qwen2-7B-instruct     | 7B    | 49.89| 51.87 | 47.12 |
> | gte-Qwen2-1.5B-instruct   | 1.5B  | 45.99| 48.03 | 43.13 |
> |         |   |  |   |   |
> | bge-m3                    | 560M  | $\underline{\text{48.23}}$ | $\underline{\text{48.78}}$ | $\underline{\text{47.45}}$ |
> | jina-embedding-v3         | 572M  | 44.94| 45.07 | 44.76 |
> | multilingual-e5-large     | 560M  | 43.78| 43.91 | 43.60 |
> | KaLM-Embedding-V2.5       | 494M  | **49.38**| **49.86** | **48.69** |
>
> **Continued on next**

---

> ### Author Response · Authors · 2025-11-21
> **(3/3) Response to Reviewer 4xdV**
>
> To assess performance in real-world industrial OOD scenarios, we conducted evaluations on two Chinese retrieval tasks:(1) customer-service FAQ retrieval, and (2) game documentation search. None of the models were trained on these datasets, ensuring a genuine OOD setting. KaLM-Embedding-V2.5 significantly outperforms Qwen3-Embedding-0.6B and even surpasses Qwen3-Embedding-8B on the game documentation search.
>
> | Model                     | Size  | Customer Service FAQ Retrieval (MRR@10) | Game Documentation Search (MRR@10) |
> |----------------------------|-------|----------------------------------------|----------------------------------|
> | Qwen3-Embedding-8B        | 8B    | **58.91**                                  | $\underline{\text{37.52}}$                            |
> | Qwen3-Embedding-0.6B      | 596M  | 54.61                                  | 33.14                            |
> | bge-m3                     | 560M  | 48.19                                  | 32.47                            |
> | gte-multilingual-base      | 305M  | 51.47                                  | 29.02                            |
> | KaLM-Embedding-V2.5        | 494M  | $\underline{\text{58.05}}$                                  | **38.24**                           |
>
> > Weakness 5: The model is not evaluated in a multilingual setting (2 high resource languages are not enough) but is compared to multilingual models. It would be great to add a multilingual evaluation
>
> > Question 1: How does the model perform on other languages than English and Chinese?
>
> Thank you for this valuable suggestion. To provide a more comprehensive multilingual evaluation, we assessed KaLM-Embedding-V2.5 on AIRBench QA, which includes seven languages: en (English), zh (Chinese), es (Spanish), fr (French), ja (Japanese), de (German), and ru (Russian). We just use a general instruction prompt for KaLM-Embedding-V2.5: *Instruct: Given a query, retrieve documents that answer the query \n Query: {query}*. This goes beyond the previously used high-resource subset (English and Chinese) and aligns with prior reviewer 4xdV's suggestions that AIRBench is more suitable for out-of-distribution evaluations.
>
> As shown in the results below, KaLM-Embedding-V2.5 demonstrates competitive performance compared to larger models (7B and 9B). For example, its average score (49.02) is nearly identical to that of the much larger gte-Qwen2-7B-instruct model (49.33). In lower-resource languages, its performance is comparable to strong multilingual embedding baselines, such as bge-m3, despite not being trained on extensive multilingual corpora. Note that Qwen3-Embedding-0.6B is not included in this table, as there are no official AIRBench evaluation results available for this model.
>
> Overall, these results demonstrate that KaLM-Embedding-V2.5 generalizes well beyond its primary English–Chinese training focus, exhibiting robust retrieval performance across a wide range of multilingual and low-resource language settings.
>
> | Models                     | Size  | MTK   | en    | zh    | es    | fr    | ja    | de    | ru    |
> |-----------------------------|-------|-------|-------|-------|-------|-------|-------|-------|-------|
> | bge-multilingual-gemma2    | 9B    | 51.77 | 46.25 | 49.34 | 60.76 | 49.69 | 60.02 | 49.77 | 54.97 |
> | gte-Qwen2-7B-instruct      | 7B    | 49.33 | 51.87 | 47.12 | 55.18 | 43.04 | 54.76 | 44.91 | 52.65 |
> | gte-Qwen2-1.5B-instruct    | 1.5B  | 45.72 | 48.03 | 43.13 | 50.26 | 40.37 | 50.04 | 41.25 | 50.73 |
> |                |   |    |  |    |   |   |   |  |   |
> | bge-m3                     | 560M  | **49.30** | $\underline{\text{48.78}}$ | $\underline{\text{47.45}}$ | $\underline{\text{53.73}}$ | **44.66**| **54.23** | **46.71** | **54.55** |
> | jina-embedding-v3          | 572M  | 45.97 | 45.07 | 44.76 | 52.19 | 39.94 | 50.11 | 43.62 | 51.70 |
> | multilingual-e5-large      | 560M  | 44.54 | 43.91 | 43.60 | 50.84 | 35.94 | 52.84 | 41.93 | 50.44 |
> | KaLM-Embedding-V2.5        | 494M  | $\underline{\text{49.02}}$ | **49.86** | **48.69** | **54.43** | $\underline{\text{43.05}}$ | $\underline{\text{52.80}}$ | $\underline{\text{46.00}}$ | $\underline{\text{52.43}}$ |

---

> > ### Comment · Reviewer_4xdV · 2025-11-21
> > **Thanks for addressing the concerns**
> >
> > Thank you for addressing most of my concerns and running more experiments, I revised my score.

---

> > > ### Author Response · Authors · 2025-11-22
> > > **Response to Reviewer 4xdV**
> > >
> > > Thank you for your thoughtful feedback and encouraging words. We sincerely appreciate your time and are glad that the revisions and experiments have addressed your concerns. We will ensure that these updates are incorporated into the final version of the paper.

---

### Official Review · Reviewer_4fB5 · 2025-10-31

**Soundness:** 3
**Presentation:** 2
**Contribution:** 3
**Rating:** 6
**Confidence:** 4

**Summary:**

This paper proposes KaLM-Embedding-V2, a 0.5B LLM-based text embedding model. The approach modifies a decoder-style architecture by removing the causal attention mask and applying mean pooling, enabling more effective bidirectional representation learning. Training follows a three-stage progressive pipeline. As a result, the model achieves SOTA on English and Chinese MTEB.

**Strengths:**

- Methodical system design. Detailed explanation of architecture, data, training all contribute
- Ablations support most claims (focal loss, hard-negative mixing, distillation components)
- Description of training data and pipeline for it’s creation

**Weaknesses:**

- The distillation stage is not fully justified, as the paper does not explore whether applying distillation earlier or interleaving phases could be equally or more effective.
- Instruction dependence remains unclear, since the paper does not evaluate performance without instructions or under instruction mismatch.
- Causal-vs-bidirectional attention choice insufficiently validated, lacking direct comparison with strong causal-mask baselines from the same model family.
- No comparison of performance to teacher (Qwen3-Embedding-8B) and no comparison to smaller (to teacher) Qwen3-4b model.

**Questions:**

Suggestions:
- it is not clear what represents different colors on figure 1
- Some citations are duplicated (MTEB, Sentence BERT, NV-Embed)
- While MTEB (v1) is larger than  MTEB (v2), MTEB v2 implements multiple fixes on existing MTEB tasks reducing estimation noise. It also removed a few faulty tasks, either due to frequent overfitting/leakage or outright bugs, e.g., SummEval has a known bug that makes its estimates unreliable (wrong). This mteb software will raise a warning about the fact. I would at least remove SummEval if not entirely replace MTEB(v1) with MTEB(v2)

---

> ### Author Response · Authors · 2025-11-21
> **(1/4) Response to Reviewer 4fB5**
>
> We sincerely thank the reviewer for the constructive comments and suggestions, which are very helpful for improving our paper. We are also grateful that you recognised the strengths of our paper. We have updated the manuscript accordingly and kindly invite the reviewer to check the revised version for details. Please kindly find point-to-point responses below.
>
> > Weakness 1: The distillation stage is not fully justified, as the paper does not explore whether applying distillation earlier or interleaving phases could be equally or more effective.
>
> Thanks for raising this valuable point. We agree that a key question is whether contrastive distillation should be applied earlier or interleaved with contrastive learning. To address this concern, we conducted additional ablations based on the pretrained checkpoints. The results show that applying KL in the first epoch improves over CL, but CL→KL outperforms KL→CL, indicating that learning coarse-grained one-hot signals first, followed by fine-grained soft signals, is most effective. While reversing the order, learning coarse-grained signals after fine-grained ones can undermine performance. KL→KL performs worse than CL→KL, as it lacks the supervised contrastive signals. Continuing training with multiple epochs (CL→CL, KL→KL) yields marginal gains, suggesting that interleaving multiple epochs is unnecessary.
> | Setting   | MTEB (cmn, v1) MTK |
> |-----------|------------------|
> | CL        | 68.15            |
> | CL → CL   | 68.31            |
> | CL → KL   | 70.72            |
> | KL        | 70.19            |
> | KL → KL   | 70.45            |
> | KL → CL   | 69.88            |
>
> > Weakness 2: Instruction dependence remains unclear, since the paper does not evaluate performance without instructions or under instruction mismatch.
>
> Thanks for raising this important question. We have added evaluations both without instructions and under instruction mismatch on MTEB (cmn, v1). Removing instructions leads to a large drop (70.93→61.19), confirming that task-specific prompts play an essential role. For instruction mismatch, each classification task in MTEB (cmn, v1) is paired with a randomly selected instruction from the remaining five tasks, and performance also decreases substantially (77.48→71.38), showing that the correct instruction matters.
> | Setting        | Mean Task | Classification | Clustering | PairClassification | Reranking | Retrieval | STS   |
> |----------------|-----------|----------------|------------|------------------|-----------|-----------|-------|
> | w/ Instruction | **70.93**     | **77.48**          | **73.09**      | **84.09**            | **66.90**     | **73.42**     | **59.80** |
> | w/o instruction| 61.19     | 72.81          | 54.08      | 65.93            | 64.40     | 64.03     | 48.84 |
>
> | Setting                | Mean Task | IFlyTek | JDReview | MultilingualSentiment | OnlineShopping | TNews  | Waimai |
> |------------------------|-----------|---------|----------|----------------------|----------------|--------|--------|
> | Matched Instruction     | **77.48**     | **56.59**   | **88.82**    | **81.26**                | **95.02**          | **53.27**  | **89.91**  |
> | Mismatched Instruction  | 71.38     | 47.65   | 85.97    | 71.78                | 89.93          | 51.14  | 81.79  |

---

> ### Author Response · Authors · 2025-11-21
> **(2/4) Response to Reviewer 4fB5**
>
> > Weakness 3: Causal-vs-bidirectional attention choice insufficiently validated, lacking direct comparison with strong causal-mask baselines from the same model family.
>
> Thanks for your valuable comment. We address this by comparing against Qwen3-Embedding-0.6B, one of the strongest causal-mask models in the same family. As shown below, KaLM-Embedding-V2.5 substantially outperforms it across both MTEB (eng/cmn, v1).
> | Models                 | MTEB (eng, v1) MTK | MTEB (eng, v1) MTY | MTEB (cmn, v1) MTK | MTEB (cmn, v1) MTY |
> |------------------------|------------------|------------------|------------------|------------------|
> | Qwen3-Embedding-0.6B   | 66.76            | 63.62            | 66.33            | 67.44            |
> | KaLM-Embedding-V2.5    | **69.33**            | **65.83**            | **70.93**            | **72.46**            |
>
> We also include an ablation that removes bidirectional attention from KaLM-Embedding-V2.5, which consistently degrades performance across MTEB (eng/cmn, v1). These results, together with comparisons to one of the strongest causal-mask baselines (i.e., Qwen3-Embedding-0.6B), validate that our bidirectional design provides a clear advantage.
> | Settings                   | MTEB (eng, v1) MTK | MTEB (eng, v1) MTY | MTEB (cmn, v1) MTK | MTEB (cmn, v1) MTY |
> |----------------------------|------------------|------------------|------------------|------------------|
> | KaLM-Embedding-V2.5        | 69.33            | 65.83            | 70.93            | 72.46            |
> | w/o Bidirectional Attention | 68.94 (-0.39)    | 65.05 (-0.78)    | 70.50 (-0.43)    | 71.95 (-0.51)    |

---

> ### Author Response · Authors · 2025-11-21
> **(3/4) Response to Reviewer 4fB5**
>
> > Weakness 4: No comparison of performance to teacher (Qwen3-Embedding-8B) and no comparison to smaller (to teacher) Qwen3-4b model.
>
> Thanks for your valuable comment. We have added Qwen3-Embedding-8B/4B results to the revised manuscript. While KaLM-Embedding-V2.5 underperforms these larger models, this gap is expected given that our model uses only 1/16 (vs. 8B) and 1/8 (vs. 4B) of their parameters.
>
> > Question 1: it is not clear what represents different colors on figure 1
>
> Thanks for your thoughtful comment, and sorry for the confusion about this figure. The colors in Figure 1 indicate models of similar parameter scales, with each scale group sharing the same color. We have clarified this explicitly in the revised manuscript.
>
> > Question 2: Some citations are duplicated (MTEB, Sentence BERT, NV-Embed)
>
> Thanks for pointing this out. We apologize for the oversight and have corrected all duplicate citations in the revised manuscript.

---

> ### Author Response · Authors · 2025-11-21
> **(4/4) Response to Reviewer 4fB5**
>
> > Question 3: While MTEB (v1) is larger than MTEB (v2), MTEB v2 implements multiple fixes on existing MTEB tasks reducing estimation noise. It also removed a few faulty tasks, either due to frequent overfitting/leakage or outright bugs, e.g., SummEval has a known bug that makes its estimates unreliable (wrong). This mteb software will raise a warning about the fact. I would at least remove SummEval if not entirely replace MTEB(v1) with MTEB(v2)
>
> Thank you for raising this important point. We fully agree that MTEB (eng, v2) addresses several known issues in MTEB (eng, v1), such as reduced noise. Following your suggestion, we conducted evaluations on MTEB (eng, v2), and we present the full results below, along with comparisons to strong baselines including Qwen3-Embedding-8B/4B/0.6B, multilingual-e5-large-instruct, bge-large-en-v1.5, and embedding-gemma-300M. The results show that KaLM-Embedding-V2.5 achieves the strongest overall performance among models of similar scale. We appreciate the reviewer 4fB5's insightful recommendation, which has helped strengthen the robustness and clarity of our evaluation. The results on MTEB (eng, v2) have been added to the revised manuscript.
> | Models                          | Size  | MTY   | MTY   | Classification | Clustering | PairClassification | Reranking | Retrieval | STS   | Summarization |
> |---------------------------------|-------|-------|-------|----------------|------------|------------------|-----------|-----------|-------|---------------|
> | Qwen3-Embedding-8B               | 8B    | 75.22 | 68.70 | 90.43          | 58.57      | 87.52            | 51.56     | 69.44     | 88.58 | 34.83         |
> | Qwen3-Embedding-4B               | 4B    | 74.60 | 68.09 | 89.84          | 57.51      | 87.01            | 50.76     | 68.46     | 88.72 | 34.39
> | |  | |   |           |        |           |     |      |   |         |
> | Qwen3-Embedding-0.6B             | 596M  | $\underline{\text{70.70}}$ | 64.88 | 85.76          | 54.05      | 84.37            | 48.18     | **61.83**     | **86.57** | $\underline{\text{33.43}}$         |
> | multilingual-e5-large-instruct   | 560M  | 65.53 | 61.21 | 75.54          | 49.89      | 86.24            | **48.74**     | 53.47     | 84.72 | 29.89         |
> | bge-large-en-v1.5                | 335M  | 65.89 | 61.87 | 78.34          | 48.01      | $\underline{\text{87.13}}$            | $\underline{\text{48.26}}$     | 55.44     | 82.79 | 33.13         |
> | embeddinggemma-300m               | 307M  | 69.67 | $\underline{\text{65.11}}$ | $\underline{\text{87.55}}$          | $\underline{\text{56.55}}$      | **87.29**            | 47.43     | 55.69     | 83.61 | **37.64**         |
> | KaLM-Embedding-V2.5               | 494M  | **71.29** | **65.31** | **90.50**          | **58.12**      | 86.63            | 47.42     | $\underline{\text{58.45}}$   | $\underline{\text{84.82}}$ | 31.21         |

---

> > ### Comment · Reviewer_4fB5 · 2025-11-21
> >
> > Thanks for the response. This has addressed at least to some extent my initial concerns. I have revised my score accordingly.

---

> > > ### Author Response · Authors · 2025-11-22
> > > **Response to Reviewer 4fB5**
> > >
> > > Thank you very much for your positive feedback and recognition of our work. We’re glad that the revisions and experiments have addressed your concerns. We will make sure to incorporate these updates into the final version of the paper.

---

### Official Review · Reviewer_U1wE · 2025-11-01

**Soundness:** 4
**Presentation:** 4
**Contribution:** 3
**Rating:** 6
**Confidence:** 4

**Summary:**

This paper presents techniques behind the KaLM text embedding model. It applies various training techniques such as focal-style re-weighing to upweigh loss on harder examples, online hard negative mixing, contrastive distillation from a bigger model, and multi-stage training.

**Strengths:**

* The final model performs well on benchmarks with fewer parameters.
* Focal-style reweighing, online hard-negative mixing and contrastive distillation are novel in the text embedding space.
* Very detailed and sounds experimental results leave few questions unanswered.

**Weaknesses:**

* The most impactful technique of focal-style reweighing is commonly used in machine learning.

**Questions:**

Where does the high performance with the small 0.5B model come from exactly? Is it the:
1) base model 2) training data 3) contrastive distillation 4) the two techniques of focal style re-weighing, hard-negative mixing.
I ask this because based on Table 5, removing both the techniques (4) would still create a strong text embedding model for 0.5B standards.

---

> ### Author Response · Authors · 2025-11-21
> **(1/3) Response to Reviewer U1wE**
>
> We sincerely thank the reviewer for the constructive comments and suggestions, which are very helpful for improving our paper. We are also grateful that you recognised the strengths of our paper. We have updated the manuscript accordingly and kindly invite the reviewer to check the revised version for details. Please kindly find point-to-point responses below.
>
> >Weakness 1: The most impactful technique of focal-style reweighing is commonly used in machine learning.
>
> Thanks for raising this thoughtful point. We agree with the reviewer `U1wE's` observation that focal-style reweighting is a technique commonly used in machine learning. However, we believe that its systematic integration within the specific context of contrastive learning for text embeddings constitutes a significant and non-trivial contribution.
>
> The technique directly addresses the ''easy sample dominance'' issue, which is particularly severe in contrastive embedding training due to the low informativeness of the enormous in-batch negatives and the necessity of a large training batch. Specifically, focal-style reweighting dynamically downweights these easy samples in the training batch, thereby forcing the model to focus on learning the more challenging samples.
>
> Crucially, this technique exhibits strong synergy with our online hard-negative mixing, where the latter generates difficult samples, and the former amplifies their learning signal.
>
> Ultimately, the impressive performance of KaLM-Embedding models is a holistic outcome of systematic optimization across well-designed model architecture, superior training techniques, and high-quality data curation, not merely the contribution of focal-style reweighting alone.

---

> ### Author Response · Authors · 2025-11-21
> **(2/3)  Response to Reviewer U1wE**
>
> > Question 1: Where does the high performance with the small 0.5B model come from exactly? Is it the:base model 2) training data 3) contrastive distillation 4) the two techniques of focal style re-weighing, hard-negative mixing. I ask this because based on Table 5, removing both the techniques (4) would still create a strong text embedding model for 0.5B standards.
>
> Thanks for raising this insightful question. We agree with the reviewer `U1wE's` point that the model exhibits relatively strong performance even with the individual ablation of the two advanced training techniques, i.e., focal style re-weighing and online hard-negative mixing, as shown in Table 5. This confirms again that the model's high performance stems from its systematic optimization of (1) well-designed model architecture, (2) superior training techniques, (3) multi-stage progressive training recipe, and (4) high-quality data curation. Each component contributes in a measurable and complementary manner:
>
> - **Model Architecture:** The model is built on Qwen2-0.5B and further enhanced by removing the causal mask to enable fully bidirectional attention. The table below shows the performance drop when bidirectional attention is ablated. The results demonstrate that embeddings generated with bidirectional attention are more effective than those generated with causal attention. It is important to clarify the role of the base model; it provides well-initialized model parameters that facilitate downstream representation learning. However, it is not the primary determinant of final embedding performance. Notably, even with a smaller and earlier-generation base model (i.e., Qwen2-0.5B), our model considerably outperforms Qwen3-Embedding-0.6B trained on a larger and more advanced Qwen3-0.6B base model. This demonstrates that while a good base model is beneficial, the performance gains mainly stem from our superior training techniques and data engineering.
> | Settings                  | MTEB (eng, v1) MTK | MTEB (eng, v1) MTY | MTEB (cmn, v1) MTK | MTEB (cmn, v1) MTY |
> |----------------------------|------------------|------------------|------------------|------------------|
> | KaLM-Embedding-V2.5       | 69.33            | 65.83            | 70.93            | 72.46            |
> | w/o Bidirectional Attention | 68.94 (-0.39)    | 65.05 (-0.78)    | 70.50 (-0.43)    | 71.95 (-0.51)    |
>
>
> - **Training Techniques:** Reviewer `U1wE` notes that removing these techniques (Table 5) yields only moderate degradation. In fact, they are essential for handling difficult samples and providing continual learning signals. The table below clearly demonstrates the performance drop when they are ablated. The notable drop of **-1.52 MTK on MTEB (cmn, v1)** after removing Focal-style Reweighting confirms its crucial role in text embedding training. On the other hand, eliminating online hard negative mixing yields smaller but consistent declines across English and Chinese, demonstrating the necessity of supplementing informative hard negatives throughout training.
> | Settings                        | MTEB (eng, v1) MTK | MTEB (cmn, v1) MTK |
> |---------------------------------|------------------|------------------|
> | KaLM-Embedding-V2.5             | 69.33            | 70.93            |
> | w/o Focal-style Reweighting     | 68.70 (–0.63)    | 69.41 (–1.52)    |
> | w/o Online Hard Negative Mixing | 68.91 (–0.42)    | 70.54 (–0.39)    |
>
> - **Training Recipe:** **The most substantial performance improvement** stems from our multi-stage training pipeline (i.e., Pretraining→Finetuning→Contrastive Distillation), particularly contrastive distillation. It moves beyond coarse-grained one-hot learning signals to fine-grained soft ones that capture nuanced differences. Contrastive distillation is crucial for embedding training, because traditional hard-label supervision treats relevance as a binary, all-or-nothing classification, i.e., positive pair=1, all others=0. This manner inaccurately reflects the document relevance and inevitably introduces false negatives. The table below shows the performance drop when the contrastive distillation stage is ablated. The performance drop is larger than any other, demonstrating contrastive distillation as one of the most crucial factors of the final high performance. Furthermore, the considerable drop observed when removing the pre-training stage, i.e., **-0.72 MTK on MTEB (eng, v1) and -0.28 MTK on MTEB (cmn, v1)**, confirms that the general-purpose representations learned during this stage are also an essential component of achieving high performance.
> | Settings                     | MTEB (eng, v1) MTK | MTEB (cmn, v1) MTK |
> |-------------------------------|------------------|------------------|
> | KaLM-Embedding-V2.5           | 69.33            | 70.93            |
> | w/o contrastive distillation  | 67.47 (–1.86)    | 68.15 (–2.78)    |
>
> **Continued on next**

---

> ### Author Response · Authors · 2025-11-21
> **(3/3) Response to Reviewer U1wE**
>
> - **Training Data:** A key factor behind the high performance is the quality and diversity of the training data, not merely its raw size. By using dataset-specific construction, hard negative mining, task-specific instructions, and example-based multi-class labeling, our data engineering fosters superior performance and generalization. To demonstrate that our performance stems from data quality and strategy, rather than raw size, the following table compares our model with a comparable baseline using different training data volumes. Our KaLM-Embedding-V2.5 achieves superior performance despite utilizing less than one-third of the finetuning data used by the Qwen3-Embedding-0.6B, verifying the effectiveness of our comprehensive data curation recipe.
> | Models                  | MTEB (eng, v1) MTK | MTEB (cmn, v1) MTK | Finetuning Dataset Size | Data Public  |
> |-------------------------|-------------------|-------------------|------------------------|--------------|
> | Qwen3-Embedding-0.6B    | 66.76             | 66.33             | 19M                    | Proprietary  |
> | KaLM-Embedding-V2.5     | **69.33**             | **70.93**             | 6M                     | **Public**       |
>
>
> In conclusion, the high performance of the KaLM-Embedding series stems from a systematic optimization throughout the entire pipeline, **especially the superior training recipe, training techniques, and data engineering**.

---

### Author Response · Authors · 2025-12-03
**General Response**

Dear Reviewers, ACs, SACs, and PCs,

We sincerely thank all reviewers (`U1wE`, `4fB5`, `4xdV`, `kXYw`) for their thoughtful comments, constructive suggestions, and valuable time, all of which have significantly improved the quality of our paper. Reviewers highlighted the strengths of our work, including new SOTA foundation embedding models, methodical system designs, comprehensive experiments, and valuable open-source contributions.

We also feel sorry for the impact of the recent OpenReview bug (Nov 27). Below, we provide a concise summary of reviewers' initial concerns and the outcomes of the rebuttal and discussion phase, during which the scores had already improved from 6,6,4,6 → 6,8,8,6, before the bug.

**Initial Concerns**

The reviewers’ initial concerns primarily focused on the following points:
- **Lack of MTEB (eng, v2) evaluation:** Reviewers noted that MTEB (Eng, v2) fixes issues in v1 and is more reliable. We have added MTEB (Eng, v2) results, further confirming the effectiveness of our model.
- **Missing Comparison to Qwen3-Embedding-8B/4B:** Reviewers pointed out the lack of comparison to Qwen3-Embedding-8B/4B. We added these results in response to the reviewers’ suggestion.
- **Need for additional multilingual evaluations:** Reviewer suggested evaluating beyond high-resource languages (Chinese and English). We added evaluations across seven languages, showing our model's strong multilingual capability.
- **Choice of initialization and necessity of pre-training:** Reviewers asked about the need for pre-training and the choice of initialization model. We added ablation studies showing that removing pre-training degrades performance. Experiments with different initialization models show that the choice of initialization model is only one of the contributing factors in building a high-quality embedding model. The superior performance of the KaLM-Embedding-V2 series arises from the synergistic effect of superior architecture design, training techniques, training recipes, and data engineering.


In response to the reviewers' comments and suggestions, we have updated the manuscript accordingly.

**Rebuttal and Discussions**

During the rebuttal, we provided extensive experiments and clarifications addressing the reviewers' concerns. The responses were submitted ~6 days before the bug. And, before the Nov 27 bug, three of the four reviewers had already responded:
- Reviewer `U1wE`: *Unfortunately, this reviewer hadn’t had time to respond yet.* **Rating remained 6.**
- Reviewer `4fB5`: "Thanks for the response. This has addressed at least to some extent my initial concerns. I have revised my score accordingly." **Rating increased 6 → 8, ~5 days before the bug.**
- Reviewer `4xdV`: "Thank you for addressing most of my concerns and running more experiments, I revised my score." **Rating increased 4 → 8, ~5 days before the bug.**
- Reviewer `kXYw`: "The empirical results are practically useful, so I maintain my original score." **Rating remained 6, ~1 day before the bug.**

**Conclusion**

We once again thank all reviewers and the AC for their time, effort, and valuable feedback, which significantly enhanced the quality of our paper.

Best regards,

Authors

---

### Meta-Review · Area_Chair_S1He · 2026-01-08

**Summary:**

The paper describes a new text embedding model of small-ish size (0.5B params) that achieves the best results on MTEB among models of <1B size, and is not very far behind SOTA much larger models. The training recipe consists of several stages, with contrastive learning on curated data and distillation from a larger model.

All reviewers were positive. The initial concerns included lack of some ablations, lack of comparison to powerful Qwen3-based embedding models, lack of MTEB v2 evaluations. There were also some more general concerns related to lack of methodological novelty.

**Reviewer Concerns:**

The authors performed a large number of additional experiments for the rebuttal, addressing most specific concerns. They added requested ablations, added MTEB v2 evaluations, and partially evaluated Qwen3-based embedding models.

Regarding Qwen3-Embedding-8B and 4B: the evaluations were added to Table 2 but only for MTEB Chinese. Why? It would be good to see the results for English as well. I assume this was due to insufficient time, and I hope the authors can include these numbers for the camera-ready. Also, please add these models to Figure 1 scatter plots.

My own main concern is that most ablations indicate only small performance drop. Most importantly, distillation alone leads to only small drop as well (Table 7), and yields a model that still wins over all <1B models in Table 2. Therefore it seems to me that the *main* message of the paper is: to get SOTA 0.5B model, distill the best 8B model. It's still cool to have it evaluated, but this is hardly a surprising result. Instead, the authors emphasize "superior training techniques and data" (see title), but actually distillation alone would do the trick *almost* as well.

Minor: MTK and MTY scores are never properly explained. The first sentence of the abstract is ungrammatical. Line 285: it is never written what V1 refers to.

**Reviewer Scores:**

The initial scores were 4/6/6/6. During rebuttal stage, two reviewers commented that they increased their scores (and indeed their specific concerns have been largely addressed). I cannot see how much they increased the scores, and I usually assume that the increase was by one step, leading to 6/6/6/8 scores. The authors wrote that one reviewer had actually increased the score by two steps, leading to 6/6/8/8, which I cannot verify. One other reviewer commented that they kept the score, the remaining reviewer would have probably kept the score too.

Either way, there is a clear consensus for accepting.

Given my own concerns (see above), I recommend acceptance but merely as a poster.

---

### Decision · Program_Chairs · 2026-01-26

Accept (Poster)